# AttentionBreaker: Adaptive Evolutionary Optimization for Unmasking Vulnerabilities in LLMs through Bit-Flip Attacks

**Sanjay Das**[*1], **Swastik Bhattacharya**[1], **Souvik Kundu**[2], **Shamik Kundu**[2], **Arnab Raha**[2], **Anand Menon**[1], **Kanad Basu**[3]

[1]*University of Texas at Dallas,* [2]*Intel Corporation,* [3]*Rensselaer Polytechnic Institute*

## Abstract

Large language models (LLMs) have significantly advanced natural language processing (NLP) yet are still susceptible to hardware-based threats, particularly bit-flip attacks (BFAs). Traditional BFA techniques, requiring iterative gradient recalculations after each bit-flip, become computationally prohibitive and lead to memory exhaustion as model size grows, making them impractical for state-of-the-art LLMs. To overcome these limitations, we propose **AttentionBreaker**, a novel framework for efficient parameter space exploration, incorporating **GenBFA**, an evolutionary optimization method that identifies the most vulnerable bits in LLMs. Our approach demonstrates unprecedented efficacy—flipping just three bits in the LLaMA3-8B-Instruct model, quantized to 8-bit weights (W8), completely collapses performance, reducing Massive Multitask Language Understanding (MMLU) accuracy from 67.3% to 0% and increasing Wikitext perplexity by a factor of $10^5$. Furthermore, AttentionBreaker circumvents existing defenses against BFAs on transformer-based architectures, exposing a critical security risk. The framework is made open sourced at: `https://github.com/TIES-Lab/attnbreaker`.

## 1  Introduction

The increasing popularity of LLMs has fundamentally expanded the capabilities of Artificial Intelligence (AI), demonstrating remarkable proficiency in generating human-like text, interpreting nuanced context, and executing complex reasoning tasks Xu et al. (2024). These advancements have not only reshaped natural language processing but have also extended AI applications into diverse fields such as computer vision and scientific research, heralding a new era of AI-driven solutions Chang et al. (2024); Xu et al. (2024). Given their pervasive use, it is critical to analyze the vulnerability of LLMs against both software-based and hardware-based threats to ensure their secure and reliable deployment Das et al. (2024).

A significant issue regarding deep learning model reliability involves threats like bit-flip attacks (BFAs), which exploit hardware vulnerabilities to corrupt memory designated for storing the model's weight parameters. Consequently, this undermines the integrity of the model performance Rakin et al. (2019); Qian et al. (2023); Kundu et al. (2024a). BFAs employ fault injection attacks such as DeepHammer Yao et al. (2020) to manipulate specific bits within DRAM, altering essential model weights to degrade functionality. Despite advancements in memory technology, recent fault injection techniques enable remote, non-physical memory manipulation, perpetuating the threat landscape for BFAs Hayashi et al. (2011); Shuvo et al. (2023). Traditional BFA methods face constraints due to iterative gradient recalculation after each bit-flip Rakin et al. (2019); Kundu et al. (2024a). While viable for smaller models, this process becomes prohibitively costly as model size grows while the effectiveness diminishes. The increasing resource demands of these recalculations lead to memory exhaustion, making conventional BFA impractical for advanced LLMs. A recent study Nazari et al. (2024) has investigated the susceptibility of transformer-based models to bit-flip attacks by extending conventional BFA techniques—albeit limited to small vision-transformers and not large models like LLMs. Furthermore, the authors proposed a potential defense against such vulnerabilities.

---

*This work is supported by the Semiconductor Research Corporation (SRC GRC Task: 3243.001) and the National Science Foundation (NSF) grant #2537757.

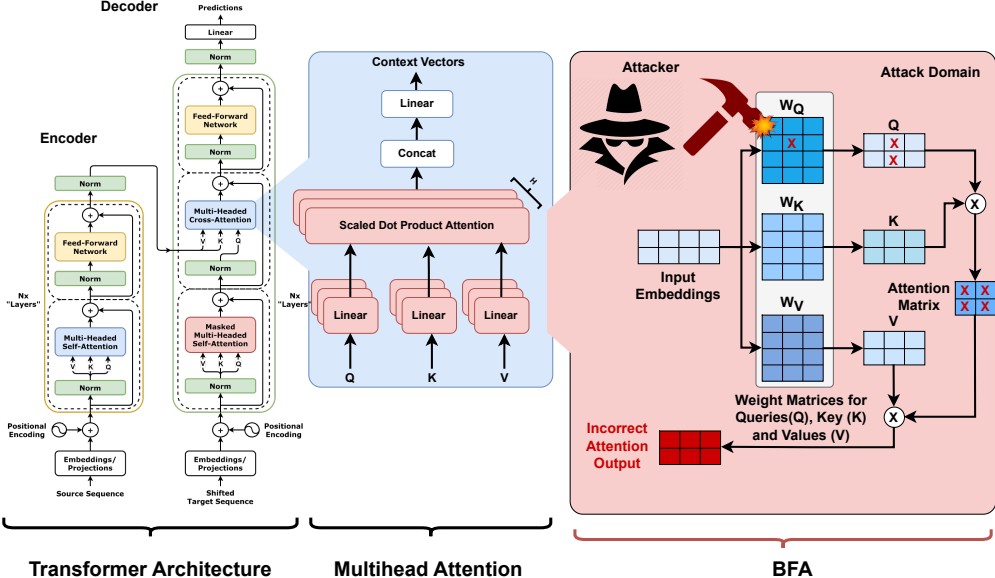

Figure 1: Bit-flip attack on transformer-based architecture.

In this paper, we introduce a fundamentally distinct methodology, marking a paradigm shift in the exploitation of LLM vulnerabilities through bit-flip attacks. In contrast to conventional BFAs, our proposed framework, **AttentionBreaker** employs a **one-shot gradient estimation** strategy, requiring a single gradient computation to aid a genetic optimization strategy, **GenBFA** that efficiently navigates and exploits the vast parameter space of LLMs. Our approach achieves unprecedented efficacy, as demonstrated on the LLaMA3-8B-Instruct W8, where only **three** strategically selected bit-flips suffice to induce catastrophic performance degradation, reducing MMLU accuracy from 67.3% to 0%. As demonstrated in Section 5.3, AttentionBreaker is more impactful compared to existing attacks Nazari et al. (2024). Furthermore, AttentionBreaker successfully subverts the defense proposed by Nazari et al. (2024), thus rendering transformer-based models vulnerable to BFAs. This fundamental redefinition of BFA methodology opens a new attack paradigm, specifically catering to LLMs, and highlights a critical reliability concern for these models deployed in mission-critical applications. Our contributions are summarized below:

- We propose **AttentionBreaker**, a novel, efficient BFA framework providing a methodical approach to navigate and exploit the parameter space of LLMs.

- We introduce **GenBFA**, as part of AttentionBreaker, a novel evolutionary algorithm for identifying critical bits in LLMs and optimizing BFAs.

- AttentionBreaker uncovers a significant vulnerability of LLMs. A mere **three** bit-flips ($\mathbf{4.129 \times 10^{-9}}$% of all bits) in LLaMA3-8B-Instruct W8, can reduce the MMLU accuracy from 67.3% to 0%, while increasing Wikitext perplexity from 12.62 to $4.72 \times 10^5$. Furthermore, AttentionBreaker has been shown to subvert existing BFA defenses on transformer-based models.

## 2 Variable Notations

This section defines the variable notations used throughout this paper, as summarized in Table 1. Consider in a model $\mathcal{M}$ with $L$-layers, parameterized by $\mathbf{W} = \{w_1, w_2, \ldots, w_n\}$, $\mathcal{L}(\mathbf{W})$ denotes the associated loss function. For simplicity, we denote the gradient of the loss function $\nabla\mathcal{L}(\mathbf{W})$ as $\nabla\mathbf{W}$ throughout this paper. To assess the sensitivity of model parameters, we define a sensitivity metric $\mathbf{S}$, with $\mathbf{S}_{\mathrm{mag}}$ and $\mathbf{S}_{\mathrm{grad}}$ capturing sensitivity based on parameter magnitudes and gradients, respectively. To balance the influence of gradients and magnitudes in the computation of a hybrid sensitivity metric, $\mathbf{S}$, a sensitivity ratio $\alpha \in [0, 1]$

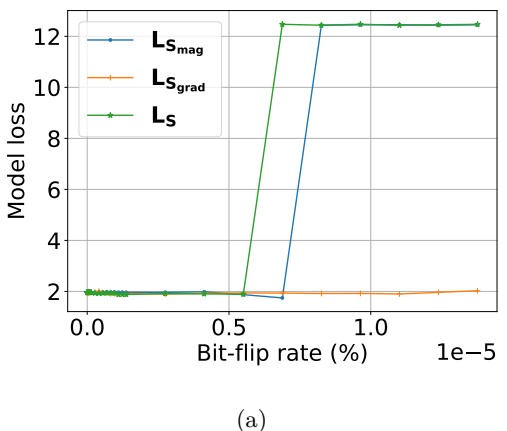 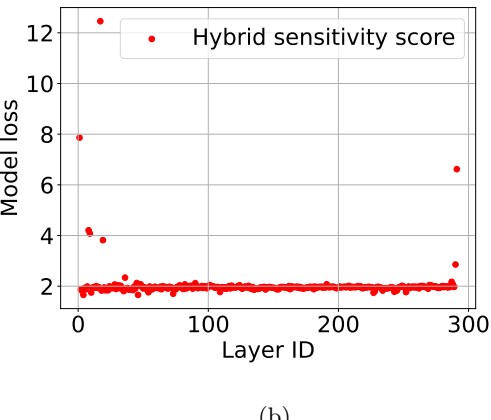

(a)                                                          (b)

Figure 2: LLaMA3-8B-Instruct W8 (a) sensitivity proxy comparison, and (b) layer sensitivity analysis on the 'Astronomy' task of MMLU benchmark.

is introduced. The attention weight matrix is represented by $\mathbf{A}$, with corresponding query and key vectors $q_i$ and $k_j$, scaled by $d_k$. The function `cardinality` is defined to evaluate the number of parameters in a given parameter set. Let $w'$ represent a perturbed version of weight $w$, and $\mathcal{R}$ denote the set of sampling rates for parameter sampling during sensitivity analysis. The normalized gradients and weights are denoted by $\nabla \mathbf{W}_N$ and $\mathbf{W}_N$, respectively. Finally, $I$ is the set of indices where bit flips are applied to perturb specific parameters.

## 3 Motivation

### 3.1 Proxy for Sensitivity

For performing BFA, it is essential to analyze model parameter sensitivity profiles independently of assumptions of robustness. In particular, parameters with larger gradients or higher magnitudes may exhibit amplified sensitivity, whereby perturbations yield disproportionately large effects on the output. In full-precision models (*e.g.*, float32), where weights $w$ span a large representable range $[-3.4 \times 10^{38}, 3.4 \times 10^{38}]$, even a single bit-flip can induce substantial perturbations, particularly for large-magnitude weights. For such weights, the sensitivity $\mathbf{S}_{\text{mag}}$ can be approximated to be proportional to the absolute magnitude ($\mathbf{S}_{\text{mag}} \propto |\mathbf{W}|$). However, gradient-derived sensitivity $\mathbf{S}_{\text{grad}}$ provides a complementary perspective, capturing the degree to which a weight change affects the model loss and thus can be approximated as proportional to the absolute gradients

Table 1: Summary of Variables and Notations.

| Variable notation | Description |
|---|---|
| $\mathcal{M}$ | The model |
| $L = \{l_1, l_2, \cdots, l_n\}$ | Layers in a model |
| $\mathbf{W} = \{w_1, w_2, \ldots, w_n\}$ | All model parameters |
| $\mathcal{L}(\mathbf{W})$ | The loss function based on the model parameters |
| $\nabla \mathcal{L}(\mathbf{W}) = \nabla \mathbf{W}$ | Gradient of the loss with respect to parameters |
| $\mathbf{S} = \{s_1, s_2, \ldots, s_n\}$ | Sensitivity of parameters to perturbations |
| $\alpha$ | Ratio of contribution to $\mathbf{S}$ by $\mathbf{W}$ and $\nabla \mathbf{W}$ |
| `cardinality` | Function to provide size of a given set |
| $\mathcal{R} = \{r_1, r_2, \cdots, r_n\}$ | Set of sampling rates |
| $I = \{i_1, i_2, \cdots, i_n\}$ | Set of indices to target for bit-flip perturbations |
| $\mathbf{W}_N, \nabla \mathbf{W}_N$ | Normalized model weights and gradients |

($\mathbf{S}_{\text{grad}} \propto |\nabla \mathbf{W}|$). Please note, that we use the standard cross-entropy loss between the model output logits and the encoded input token IDs evaluated independently for each downstream task using the corresponding task-specific test data (e.g., MMLU for language models) Mao et al. (2023). Therefore, a hybrid sensitivity metric is proposed one that considers both magnitude and gradient influences to capture the sensitivity profile holistically, and is therefore expressed as:

$$\mathbf{S} = \alpha \cdot |\nabla \mathbf{W}| + (1 - \alpha) \cdot |\mathbf{W}| \tag{1}$$

Where $\alpha$ is a tunable parameter balancing the importance of magnitude and gradient. Figure 2a illustrates our experiments with sensitivity proxies based on gradients ($|\nabla \mathbf{W}|$), magnitudes ($|\mathbf{W}|$), and the hybrid sensitivity metric $\mathbf{S}$ with top-k bit flips on a LLaMA3-8B-Instruct W8 model. We observe that bit-flip perturbations guided by the hybrid metric $\mathbf{S}$ (green) induce a comparable loss increase with fewer bit flips than both $\mathbf{S}_{\text{grad}}$ (orange) and $\mathbf{S}_{\text{mag}}$ (blue).

### 3.2 Layer Sensitivity Analysis

Identifying critical bits in LLMs is challenging due to their vast parameter space. However, finding a sensitive layer is more manageable because there are fewer layers than total parameters. To quantify layer sensitivity, we choose top-$k$ bit-flips in each layer using a sampling rate $r$, guided by the hybrid sensitivity score $\mathbf{S}$, and observe the resulting model loss $\mathcal{L}$. $k$ is calculated as:

$$k = \texttt{cardinality}(\mathbf{W}^{(l)}) \times \frac{r}{100} \tag{2}$$

where $\mathbf{W}^{(l)}$ represents parameters in layer $l$. The losses vary significantly across layers, as illustrated in Figure 2b on the LLaMA3-8B W8 model. Some layers show much greater sensitivity than others, indicating that focusing on these layers can significantly reduce the search space.

### 3.3 Weight Subset Optimization

The subset of weights in a victim LLM that can be perturbed may be sufficiently large (*e.g.,* 5,872 for the LLaMA3-8B-Instruct W8 model), making an effective adversarial attack impractical. This requires further optimization to reduce the subset to fewer weights and/or bits for a feasible attack. Solving this problem through exhaustive approaches is computationally expensive due to the vast solution space. To address this challenge, we propose a heuristic optimization technique (Section 4.2.2) inspired by evolutionary strategy, which simultaneously optimizes for higher model loss (to maximize attack effectiveness) and fewer bit-flips (to enhance attack efficiency). As explained in Section 4.2.2, this approach successfully reduced the required bit-flips *from 5,872 to just 3*, achieving an approximate reduction of $\mathbf{2 \times 10^3}$.

## 4 Proposed AttentionBreaker Methodology

### 4.1 Threat Model

Machine Learning as a Service (MLaaS) provides users with access to high-performance computing resources, thereby enabling the deployment of LLMs. However, this cloud-based paradigm introduces critical security risks, particularly when inferences and model executions share server hardware components with potentially untrusted processes. We categorize the threats into four levels based on the extent of knowledge an attacker can extract from the victim, detailed in the appendix (Section A.1).

In this work, we consider the white-box threat model, where the adversary has full access to the model's architecture and parameters, enabling precise and informed manipulation of parameters. Within this setting, we focus on untargeted attacks, which aim to induce graceless degradation in overall model performance rather than targeting specific outputs. Such attacks are particularly insidious, as they compromise accuracy across a broad range of inputs while avoiding distinct failure patterns, making them more challenging to detect and defend against than targeted attacks Özdenizci & Legenstein (2022); Qian et al. (2023); Nazari

et al. (2024). We employ the standard cross-entropy loss Mao et al. (2023), computed between the output logits of the LLM and the corresponding ground-truth token IDs, as a measure of model performance. A lower cross-entropy loss indicates that the model assigns higher likelihood to the correct tokens and thus performs better. Our objective is to identify the most critical subset of parameter bits such that, when flipped, they cause a substantial increase in the cross-entropy loss. This increase directly translates into severe degradation of model performance, thereby highlighting the vulnerability of the model to targeted bit-flip manipulations.Therefore, AttentionBreaker strategically employs an untargeted attack to maximize disruption while evading traditional defense strategies. *This attack also fundamentally differs from denial of service attacks, which aim to overwhelm system resources to render services unavailable.* In our case, the adversary subtly manipulates model parameters to degrade performance without causing a complete system shutdown. By focusing on untargeted BFAs in a white-box setting, our work aims to highlight the critical need for robust defense mechanisms to ensure the reliability and security of transformer-based models in shared computational environments.

## 4.2 Proposed Attack Framework

Here, we explain in detail the proposed AttentionBreaker framework, which comprises two structured steps: (1) layer ranking based on their sensitivity profiles and (2) weight subset optimization to identify the model's most critical parameters for an efficient attack.

### 4.2.1 Layer Ranking

For ranking LLM layers, we begin by quantifying the sensitivity of each layer's parameters. For this purpose, each weight $w_i$ in a layer is evaluated through a sensitivity scoring function as defined in Equation 1.

---

**Algorithm 1** Layer Ranking

---

**Require: W**, $\nabla \mathbf{W}$, $\alpha$, sub-sampling % $r$, # of top layers $n$
1: Initialize sensitivity score set: $\mathcal{L}_{sens} = [\ ]$
2: **function** BFLIP($\mathbf{W}^{(l)}$, $pos$, $I$)
3:     $\mathbf{W}^{(l)}_{orig} \leftarrow \mathbf{W}^{(l)}$
4:     **for all** $i \in I$ **do**
5:         $w^{(l)}_i \leftarrow w^{(l)}_i \oplus (1 \ll pos)$
6:     **end for**
7:     $\mathcal{M} \leftarrow$ updateModel($\mathcal{M}, \mathbf{W}^{(l)}$)
8:     $\mathcal{L} \leftarrow$ ComputeLoss($\mathcal{M}$)
9:     $\mathcal{M} \leftarrow$ updateModel($\mathcal{M}, \mathbf{W}^{(l)}_{orig}$)
10:    **return** $\mathcal{L}$
11: **end function**
12: **for all** $l \in L$ **do**
13:    $k \leftarrow r *$ cardinality($\mathbf{W}^{(l)}$)/100
14:    $\mathbf{S}^{(l)} \leftarrow \alpha|\nabla \mathbf{W}^{(l)}| + (1-\alpha)|\mathbf{W}^{(l)}|$
15:    $I^{(l)}_{hybrid} \leftarrow$ Top$k$Index($\mathbf{S}^{(l)}$)
16:    $\mathcal{L}^{(l)} \leftarrow$ BFLIP($\mathbf{W}^{(l)}, pos, I^{(l)}_{hybrid}$)
17:    LSScore$^{(l)} \leftarrow$ sgn$\left(\mathcal{L}^l - \mathcal{L}_{th}\right) \cdot \frac{\mathcal{L}^l}{k}$
18:    Push [LSScore$^{(l)}, l$] to $\mathcal{L}_{sens}$
19: **end for**
20: $\mathcal{L}_{sens} \leftarrow$ SORT($\mathcal{L}_{sens}$)
21: $\mathcal{L}_{top} \leftarrow$ Top$n(\mathcal{L}_{sens})$
22: $\mathbf{W}_{sub} \leftarrow [l, I^{(l)}_{hybrid}]$ extracted from $\mathcal{L}_{top}$
**Ensure:** $\mathcal{L}_{sens}, \mathbf{W}_{sub}$

---

Please note that for this scoring we only use the most significant bit (MSB) flips as experimental evaluation consistently demonstrated that flipping the MSB caused the most impact. Using these sensitivity scores, weights are aggregated and ranked in descending order to form a prioritized list of the most critical weights in each layer. The top-$k$ weights within each layer are identified using Equation 2. To maximize the impact of bit-flips, perturbations are applied by targeting the most significant bits (MSBs) to induce the highest deviation for each selected weight. After applying these perturbations, we compute the model's loss, $\mathcal{L}^{(l)}$, which reflects the sensitivity of the layer. In order to identify layers with heightened sensitivity while requiring a minimal number of bit-flips, we define a layer sensitivity score (LSScore) function. First, we define a sign function sgn$\left(\mathcal{L}^{(l)} - \mathcal{L}_{th}\right)$ based on whether the model loss $\mathcal{L}^{(l)}$ evaluated after bit-flipping the weights in layer $l$ meets a predefined critical loss threshold $\mathcal{L}_{th}$:

$$\text{sgn}\left(\mathcal{L}^{(l)} - \mathcal{L}_{th}\right) = \begin{cases} +1, & \text{if } \mathcal{L}^{(l)} - \mathcal{L}_{th} \geq 0, \\ -1, & \text{if } \mathcal{L}^{(l)} - \mathcal{L}_{th} < 0. \end{cases} \quad (3)$$

Using this sign function, we define LSScore as:

$$\text{LSScore}^{(l)} = \text{sgn}\left(\mathcal{L}^{(l)} - \mathcal{L}_{th}\right) \cdot \frac{\mathcal{L}^{(l)}}{k} \quad (4)$$

A layer-wise bit-flip sensitivity profile is created by iterating through each layer to record their sensitivity scores. This profile aids in identifying highly sensitive layers.

Algorithm 1 presents a systematic approach for conducting layer sensitivity analysis. The layer sensitivity analysis itself begins by iterating through each layer, assessing the model's sensitivity to perturbations in each (*lines 12-19*). Sensitivity scores $\mathbf{S}^{(l)}$ for the weights in layer $l$ are then computed using Equation 1 (*line 14*). Next, the top-$k$ indices are obtained using sensitivity scores $\mathbf{S}^{(l)}$ with the function `TopkIndex`. These indices, $I_{hybrid}^{(l)}$, are identified from $\mathbf{S}^{(l)}$ (*line 15*). These indices, along with the weights $\mathbf{W}^{(l)}$ and bit position *pos*, are passed to the `BFLIP` function to compute the model loss $\mathcal{L}_r^{(l)}$ and the corresponding layer sensitivity score `LSScore` (*lines 16-17*). The resulting score `LSScore`$^{(l)}$ is then stored in $\mathcal{L}_{sens}$ (*line 18*). This process continues until all layers have been processed, yielding the final sensitivity score set $\mathcal{L}_{sens}$, which is then sorted, and top-$n$ layers are chosen (*lines 20-21*). Finally, the top-$n$ most critical layers and their corresponding weight indices $I_{hybrid}^{(l)}$ are extracted to generate a critical weight subset $\mathbf{W}_{sub}$ (*line 22*), which along with the sensitivity scores $\mathcal{L}_{sens}$ are provided as the output.

### 4.2.2 Weight-set Optimization - GenBFA

After identifying the most sensitive layer and the corresponding weight subset, further refinement is essential since LLMs typically encompass billions of parameters, and the identified weight subset may necessitate targeting thousands or even millions of parameters. For example, the LLaMA-3-8B W8 has 8 billion parameters, and the weight subset corresponding to the most sensitive layer has $5,872$ parameters. Consequently, an attack with this identified weight set could be prohibitive from an attacker's perspective. Therefore, it is imperative to identify a smaller subset of parameters capable of achieving comparable degradation in model performance. To address this objective, we design a novel algorithm inspired by evolutionary strategies, **GenBFA**, tailored for optimizing the parameter set to identify a minimal yet highly critical subset of weights. GenBFA maximizes the impact-to-size ratio of the selected parameters, thereby enhancing the efficiency and effectiveness of the AttentionBreaker attack on LLMs.

**Objective Function and Fitness Evaluation:** To formalize the weight subset optimization problem, let us define an initial weight subset $\mathbf{W}_{sub}$ and a target model loss threshold $\mathcal{L}_{th}$. The objective is to identify a subset $\mathbf{W}_{S_j} \subset \mathbf{W}_{sub}$ such that bit-flipping the weights in this subset maximizes the model's loss while minimizing the subset's `cardinality`, which reflects the number of bit-flips required to achieve a corresponding model loss, thus providing a measure of the attack's efficiency. Each candidate solution $\mathbf{W}_{S_j} = \{w_{j_1}, w_{j_2}, \ldots\}$ is therefore defined as a subset of $\mathbf{W}_{sub}$. For each candidate $\mathbf{W}_{S_j}$, we compute a fitness value $f(\mathbf{W}_{S_j})$ as follows.

$$f(\mathbf{W}_{S_j}) = \text{sgn}\left(\mathcal{L}_{S_j} - \mathcal{L}_{th}\right) \cdot \frac{\mathcal{L}_{S_j}}{\texttt{cardinality}(\mathbf{W}_{S_j})} \tag{5}$$

**Population Initialization:** To identify the final set of critical parameters, we define a solution set (also referred to as the population) with the objective of minimizing its cardinality while ensuring that the fitness score remains above a predefined threshold. The population $\mathcal{P}$, comprises of $m$ candidate solutions, defined as $\mathcal{P} = \{\mathbf{W}_{S_1}, \mathbf{W}_{S_2}, \ldots, \mathbf{W}_{S_m}\}$. The initial solution $\mathcal{P}_0$ is set as the weight subset $\mathbf{W}_{sub}$, and each subsequent solution $\mathcal{P}_j$ is created by applying a mutation operation to $\mathcal{P}_0$ (`Mutate`($\mathcal{P}_0$)).

**Mutation Operation:** During the mutation process, each solution $\mathcal{P}_j$ is assigned a distinct reduction rate $p_j$, which is randomly selected such that $p_j \leq \mu$, where $\mu$ represents the mutation rate. The parameter $\mu$ serves as an upper bound on the reduction of solution cardinality, thereby facilitating enhanced exploration of the solution space. The reduction rate $p_j$ then governs the probability of removal for each weight in $\mathcal{P}_j$. Formally, for each weight $w_{j_i} \in \mathcal{P}_j$, the mutated weight $w'_{j_i}$ is defined as:

$$w'_{j_i} = \begin{cases} \emptyset, & \text{with probability } p_j, \\ w_{j_i}, & \text{otherwise.} \end{cases} \tag{6}$$

In this context, "mutation" specifically refers to reducing the solution set's cardinality. This reduction-driven mutation is designed to identify compact subsets that either preserve or improve model loss relative to the original subset. By systematically exploring minimal yet sufficient configurations, this strategy facilitates

efficient traversal of the solution subspace and the identification of subsets that sustain the desired adversarial effect on model performance.

Therefore, mutation is performed by randomly reducing the size of the candidate subset according to a variable reduction rate (ranging between 1–10% and chosen randomly during each experiment), thereby introducing evolutionary pressure variably toward compact, high-impact solutions.

---

**Algorithm 2** Weight Set Optimization - GenBFA

**Require: W**, $\mathbf{W}_{sub}$, $\mathcal{L}_{th}$, population size $m$, max generations $g$, mutation rate $\mu$, no-improvement threshold $\mathcal{N}$
 1: Initialize solution space $\mathbf{SS} = [\ ]$, solution set $\mathcal{P} = [\ ]$, fitness scores $\mathbf{F} = [\ ]$, no-improvement counter $c = 0$
 2: $l, \mathbf{SS} \leftarrow \mathbf{W}_{sub}[0], \mathbf{W}_{sub}[1]$
 3: $\mathcal{P} \leftarrow \texttt{InitializePopulation}(\mathbf{SS}, \mu)$
 4: **for** $t = 1, 2, \ldots, g$ **do**
 5:     **for** $j = 1, 2, ..., m$ **do**
 6:         $\mathbf{W}^{(l)} = \texttt{GetParams}(\mathbf{W}, l)$
 7:         $I_{hybrid}^{(l)} \leftarrow \mathcal{P}_j$
 8:         $\mathcal{L}_j \leftarrow \texttt{BFLIP}(\mathbf{W}^{(l)}, pos, I_{hybrid}^{(l)})$
 9:         $f \leftarrow \text{sgn}\,(\mathcal{L}_j - \mathcal{L}_{th}) * \frac{\mathcal{L}_{S_j}}{\texttt{cardinality}(\mathbf{W}^{(l)})}$
10:         Push $f$ to $\mathbf{F}$
11:     **end for**
12:     $\mathcal{P} \leftarrow \texttt{SortSolutions}(\mathcal{P}, \mathbf{F})$
13:     $\mathcal{P}_{best}^{old} \leftarrow \mathcal{P}_{best}$
14:     $\mathcal{P}_{best} \leftarrow \texttt{SelectBestSolution}(\mathcal{P})$
15:     $\texttt{CheckTerminate}(\mathcal{P}_{best}, \mathcal{P}_{best}^{old}, c, \mathcal{N})$
16:     $\mathcal{P}_{new} \leftarrow [\mathcal{P}_{best}, \texttt{Mutate}(\mathcal{P}_{best})]$
17:     **while** $\texttt{len}(\mathcal{P}_{new}) < m$ **do**
18:         $\mathcal{P}_{p1}, \mathcal{P}_{p2} \leftarrow \texttt{Selection}(\mathcal{P}, \mathbf{F})$
19:         $\mathcal{P}_{o1}, \mathcal{P}_{o2} \leftarrow \texttt{Crossover}(\mathcal{P}_{best}, \mathcal{P}_{p1}, \mathcal{P}_{p2})$
20:         $\mathcal{P}_{o1}, \mathcal{P}_{o2} \leftarrow \texttt{Mutate}(\mathcal{P}_{o1}, \mu), \texttt{Mutate}(\mathcal{P}_{o2}, \mu)$
21:         Push $\mathcal{P}_{o1}$ and $\mathcal{P}_{o2}$ to $\mathcal{P}_{new}$
22:     **end while**
23:     $\mathcal{P} \leftarrow \mathcal{P}_{new}$    ▷ Population for next generation
24: **end for**
**Ensure:** $\mathcal{P}_{best}$          ▷ Return best solution

---

**Elitism and Selection:** At each iteration ($t$) of the optimization procedure, all solutions in $\mathcal{P}$ are sorted by their fitness values:

$$\mathcal{P}_{best} = \mathbf{W}_{S_{best}} = \arg\max_{\mathcal{P}_j \in \mathcal{P}} f(\mathcal{P}_j) \qquad (7)$$

Here, $\mathbf{W}_{S_{best}}$ or $\mathcal{P}_{best}$ represents the solution with the highest fitness (*i.e.*, most efficient in degrading model performance). Elitism dictates that $\mathcal{P}_{best}$ is carried forward unchanged into the next generation, ensuring the highest-quality solution is preserved. For the remaining solutions, a tournament selection strategy is employed to choose pairs of solutions as parents for crossover. For selecting each parent, this involves choosing two random solutions $\mathcal{P}_a, \mathcal{P}_b \in \mathcal{P}$, and selecting the one with better fitness:

$$\mathcal{P}_p = \arg\max\{f(\mathcal{P}_a), f(\mathcal{P}_b)\} \qquad (8)$$

This strategy is chosen over a random strategy due to its capability to balance exploration and exploitation in the search space, thereby expediting convergence toward an optimal solution.

**Crossover Operation:** To generate new solutions, two-parent solutions, $\mathcal{P}_{p1}$ and $\mathcal{P}_{p2}$, are selected, followed by a crossover operation executed with probability $p_c$. During crossover, the current best solution $\mathcal{P}_{\text{best}}$ is integrated to propagate its beneficial traits forward. Two offsprings, $\mathcal{P}_{o1}$ and $\mathcal{P}_{o2}$, are produced by combining $\mathcal{P}_{\text{best}}$ with each parent $\mathcal{P}_{p1}$ and $\mathcal{P}_{p2}$ independently. For offspring $\mathcal{P}_{o1}$, each gene $g_i$ is chosen stochastically from either $\mathcal{P}_{p1}$ or $\mathcal{P}_{\text{best}}$. Similarly, offspring $\mathcal{P}_{o2}$ is created by selecting each gene $g_i$ stochastically from $\mathcal{P}_{p2}$ or $\mathcal{P}_{\text{best}}$.

$$\mathcal{P}_{o1}, \mathcal{P}_{o2} = \{g_{\mathcal{P}_{p1}}^i \text{ or } g_{\mathcal{P}_{\text{best}}}^i\}, \{g_{\mathcal{P}_{p2}}^i \text{ or } g_{\mathcal{P}_{\text{best}}}^i\} \quad \forall i \qquad (9)$$

If crossover does not occur (*i.e.*, with probability $1 - p_c$), the offspring remain identical to the parents, such that $\mathcal{P}_{o1} = \mathcal{P}_{p1}$ and $\mathcal{P}_{o2} = \mathcal{P}_{p2}$.

**Convergence and Termination:** The process iterates over generations indexed by $t = 1, 2, \ldots, g$, where $g$ is the predefined maximum number of iterations. The algorithm terminates when either the iteration limit $g$ is reached ($t = g$) or a predefined no-improvement limit $\mathcal{N}$ is met, ensuring computational efficiency. During our experiments, the number of generations is capped at 100, with early stopping applied whenever no improvement is observed for a fixed number (default 5) of iterations. The final output is the weight subset $\mathbf{W}_{opt} \subset \mathbf{W}_{sub}$, where $\mathbf{W}_{opt} = \arg\max_{\mathcal{P}_j \in \mathcal{P}} f(\mathcal{P}_j)$. $\mathbf{W}_{opt}$ represents the critical weights required to achieve the desired model loss degradation.

Algorithm 2 outlines the optimization process, starting with model weights $\mathbf{W}$, a target loss threshold $\mathcal{L}_{th}$, and parameters for population size $m$, maximum iterations $g$, crossover probability $p_c$, and mutation rate $\mu$.

It initializes the solution space **SS**, population $\mathcal{P}$, and fitness scores **F** (*line 1*), followed by generating the attacked layer $l$ and initial solution space from $\mathbf{W}_{sub}$ (*line 2*). The initial solution $\mathcal{P}_0$ ($=\mathbf{W}_{sub}$) is used to create a population by applying $m-1$ mutations, resulting in $\mathcal{P}$ (*line 3*). At each iteration $t$, solutions are evaluated using the BFLIP function, which computes model loss $\mathcal{L}_j$, and the fitness score is determined based on $\mathcal{L}_j$, $\mathcal{L}_{th}$, and the cardinality of $\mathbf{W}^{(l)}$ using Equation 5 (*lines 6-10*). Solutions are ranked by fitness (*line 12*). The current best solution is tallied against the previous best solution for capturing improvement and checking for termination using CheckTerminate function (*lines 13-15*). To maintain elitism, the best solution $\mathcal{P}_{best}$ is carried over to the next generation $\mathcal{P}_{new}$ (*ilne 16*). A second solution is created by mutating $\mathcal{P}_{best}$, and the remaining population is filled by generating offspring through crossover and mutation (*lines 17-20*). Crossover selects parent pairs via tournament selection (*line 18*), generates offspring (*line 19*), and applies mutation (*line 20*), which are then added to $\mathcal{P}_{new}$ (*line 21*). The new population $\mathcal{P}_{new}$ replaces $\mathcal{P}$ (*line 23*), and the process repeats until either $g$ generations or the no-improvement threshold $\mathcal{N}$ is reached. The algorithm outputs $\mathcal{P}_{best}$ as the optimal subset of critical weights, targeting for BFA.

Table 2: AttentionBreaker evaluation on various models and datasets.

| Model | Number of Parameters | Existing Research Nazari et al. (2024) (# of bit-flips ) | Attention Breaker (# of bit-flips) | Improvement | Benchmark performances (Before attack / After Attack) | | | | |
|---|---|---|---|---|---|---|---|---|---|
| | | | | | Perplexity | Accuracy (%) | | | |
| | | | | | WikiText-2 | MMLU | LM Harness | VQAv2 | TextVQA |
| LLaMA3-8B-Instruct-W8 | 8.03 B | 587202 | 3 | $195,734\times$ | $12.14/4.7\times10^5$ | 67.3/0 | 72.6/0 | NA | NA |
| LLaMA3-8B-Instruct-W4 | | 587202 | 28 | $20,971\times$ | $12.60/3.8\times10^4$ | 60.3/0 | 61.2/0 | NA | NA |
| LLaVA1.6-Mistral-7B-W8 | 7.97 B | 419430 | 15 | $27,962\times$ | NA | NA | NA | 81.4/0 | 64.2/0 |
| LLaVA1.6-Mistral-7B-W4 | | 838860 | 53 | $15,828\times$ | NA | NA | NA | 75.6/0 | 60.5/0 |
| BitNet-large | 729 M | 314572 | 45861 | $7\times$ | $17.21/5.5\times10^4$ | 22.1/0 | 23.2/0 | NA | NA |
| Phi-3-mini-128k-Instruct-W8 | 3.82 B | 1258291 | 4 | $314,573\times$ | $11.82/3.7\times10^4$ | 69.7/0 | 72.3/0 | NA | NA |
| Phi-3-mini-128k-Instruct-W4 | | 2516 | 4 | $629\times$ | $11.94/3.6\times10^4$ | 66.1/0 | 71.7/0 | NA | NA |

# 5 Experimental results

## 5.1 Experimental Setup

For our experiments, we utilized a wide range of models, comprising of LLaMA3-8B-Instruct Touvron et al. (2023); Dubey et al. (2024), Phi-3-mini-128k-Instruct Abdin et al. (2024) and BitNet Ma et al. (2024). The evaluation of LLMs was conducted using standard benchmarks like MMLU Gema et al. (2024) and the Language Model (LM) Evaluation Harness Gao et al. (2024), both of which test the models' ability to reason and generalize across a variety of tasks. We extended our evaluation to include a Vision-Language Model (VLM), LLaVA1.6 Liu et al. (2023). The performance of this VLM was assessed using the VQAv2 Nguyen et al. (2023) and TextVQA Singh et al. (2019) benchmarks. To demonstrate the versatility of AttentionBreaker, we utilize several quantization formats, including the standard signed 8-bit integer (INT8), normalized float 4-bit (NF4), and an advanced 1.58-bit precision format (where weights are ternary $\{-1,0,1\}$), implemented using BitNet Ma et al. (2024). Please note that all these models are weight-only quantized, and we will use notations W8, W4, and W1.58, respectively.

We employ perplexity and accuracy as evaluation metrics across multiple benchmarks. Perplexity Hu et al. (2024) assesses a model's predictive capability, defined as the exponential of the average negative log-likelihood of a sequence. Accuracy measures the proportion of correct predictions, providing a direct evaluation of model performance.

$$\text{Perplexity} = \exp\left(-\frac{1}{N}\sum_{i=1}^{N}\log P(x_i)\right) \tag{10}$$

where $N$ is the number of words, and $P(x_i)$ is the probability assigned to the $i$-th word by the model. A lower perplexity indicates better performance. Accuracy on benchmarks is another critical metric, especially for tasks such as classification Gupta et al. (2024). It is the ratio of correctly predicted instances to the total

Table 3: AttentionBreaker experiment parameters values.

| Parameter | Value |
|---|---|
| Top-n Layers | 1-5 (default = 1) |
| Top-k Weights | 0.001%-10% of layer weights (default = 0.5%) |
| Initial Number of Solutions | 20-100 (default = 50) |
| Maximum Generations | 20-100 (default = 50) |
| Number of Solutions in each GenBFA generation | 40 |
| Number of generations allowed with no-improvement | 5 |
| Mutation Rate | 0.1 |
| Crossover Probability | 0.9 |

instances.

$$\text{Accuracy} = \frac{1}{N} \sum_{i=1}^{N} \mathbb{I}(\hat{y}_i = y_i) \tag{11}$$

Where $N$ is the total number of tasks. $\hat{y}_i$ is the predicted label for the $i$-th task. $y_i$ indicates the true label for the $i$-th task. $\mathbb{I}$ is the indicator function that returns 1 if the predicted label matches the true label and 0 otherwise.

Table 3 lists the parameter values used in our AttentionBreaker experiments. These values are not fixed and may be adjusted or refined by the user to achieve improved convergence.

## 5.2 Targeted Transformer Modules

Transformer-based LLMs comprise self-attention projection layers (query, key, value, and output), MLP projection layers (gate, up, and down), token embedding, layer normalization, and the LM head. Parameter distribution typically allocates 60–70% to MLP projections, 10–20% to self-attention projections, $\sim 5$ˇ7% each to token embeddings and the LM head, and $< 0.01\%$ to normalization layers. Since quantization applies only to the MLP and self-attention projections (86.9% of the parameters), AttentionBreaker exclusively targets these layers.

## 5.3 Results and Analysis

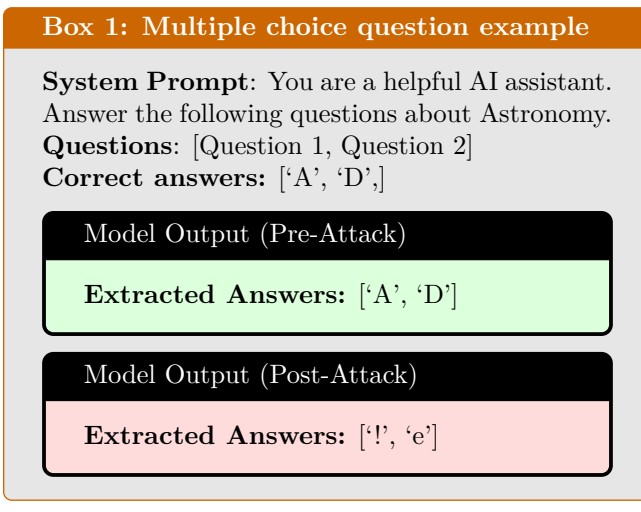

Box 1: Multiple choice question example

**System Prompt**: You are a helpful AI assistant. Answer the following questions about Astronomy.
**Questions**: [Question 1, Question 2]
**Correct answers**: ['A', 'D',]

Model Output (Pre-Attack)

**Extracted Answers**: ['A', 'D']

Model Output (Post-Attack)

**Extracted Answers**: ['!', 'e']

This section presents the experimental results validating the effectiveness of the proposed AttentionBreaker attack. Box 1 shows an example of how the LLaMA3-8B-Instruct W8 model's outputs are degraded by AttentionBreaker, yielding incoherent and incorrect answers. Section A.4 discusses additional examples.

### 5.3.1 Efficiency of AttentionBreaker

This section shows AttentionBreaker's effectiveness in degrading LLM performance across benchmarks (see Table 2) and compares it to existing research Nazari et al. (2024). The first two columns list model attributes: name, quantization scheme, and parameter count. Columns three and four specify the bit-flips needed by each approach for full performance collapse. The fifth column measures AttentionBreaker's efficiency improvement over Nazari et al. (2024). The final five columns present benchmark results, including WikiText-2 perplexity and model accuracy on MMLU, LM Harness, VQAv2,

Table 4: Performance Comparison between AttentionBreaker and conventional BFA Rakin et al. (2019).

| Model | # Bit-flips | Accuracy before attack (%) | Accuracy after BFA (%) | Accuracy after AttentionBreaker (%) |
|---|---|---|---|---|
| LLaMA3-8B-Instruct-W8 | 1000 | 67.3 | 67.3 | 0.0 |
| Phi-3-mini-128k-Instruct-W8 | 1000 | 69.7 | 69.7 | 0.0 |

Table 5: Timing overhead Comparison between AttentionBreaker and conventional BFA.

| Model | # Bit-flips | Time overhead (Hours) | |
|---|---|---|---|
| | | BFA | AttentionBreaker |
| LLaMA3-8B-Instruct-W8 | 1000 | $\sim$20 | $\sim$2 |
| LLaMA3-8B-Instruct-W4 | 1000 | $\sim$20 | $\sim$2 |

and TextVQA, both before and after the attack. "NA" indicates when benchmark performance does not apply to the model.

Table 2 highlights AttentionBreaker's efficiency in degrading model accuracy with minimal bit-flips (as few as 3), while Nazari et al. (2024) requires up to $1.3 \times 10^6$ bit-flips for causing similar performance degradation. AttentionBreaker reduces LLaMA3-8B-Instruct W8's MMLU accuracy from 67.3% to 0% with just three bit-flips ($4.129 \times 10^{-9}$% bit-flip rate) in the MLP down projection of the second decoder block. Additionally, perplexity increases from 12.14 to $4.72 \times 10^5$ post-attack. In contrast, Nazari et al. (2024) requires $5.9 \times 10^5$ bit-flips (0.1% bit-flip rate) for similar degradation, demonstrating that AttentionBreaker is $1.9 \times 10^5$ times more efficient. A similar trend is observed for Phi-3-mini-128k-Instruct and LLaVA1.6-Mistral-7B W8 models. The critical layers affected are the self-attention key, value projection, and MLP down projection in the initial three and final three decoder blocks.

### 5.3.2 Comparison across Quantization Levels

Figure 3 compares AttentionBreaker with Nazari et al. (2024) on LLaMA3-8B-Instruct W8, W4, and BitNet W1.58. The green/blue lines indicate post-attack model loss, while the orange/brown lines show MMLU accuracy. AttentionBreaker consistently reduces accuracy to 0, demonstrating its effectiveness through a rapid loss increase, as shown in Figure 3a for LLaMA3-8B-Instruct 8-bit quantized model. Figure 3b examines AttentionBreaker on 4-bit quantized LLaMA3-8B-Instruct, where just **28 bit-flips** increase perplexity from 12.60 to $3.8 \times 10^4$, dropping the MMLU score from 60.3% to 0. In contrast, Nazari et al. (2024) requires $5.9 \times 10^5$ bit-flips for the same effect. Similar trends are observed for Phi-3-mini-128k-Instruct and LLaVA1.6-Mistral-7B W4. For the ultra-low precision model BitNet W1.58 Ma et al. (2024), AttentionBreaker degrades the performance to 0% by flipping 45811 bits ($6.27 \times 10^{-5}$ % of all bits) as shown in Figure 3c. In comparison, Nazari et al. (2024) requires $6.8\times$ more bit-flips. This disparity in number of bit-flips required for BitNet in comparison to LLaMA3-8B-Instruct-W8 and -W4 can be attributed to the unique architecture and quantization strategies employed in BitNet. Unlike W8 or W4 models, which have higher precision and therefore higher weight representation range, BitNet employs 1.58-bit quantization (ternary representation -1, 0, +1), significantly limiting the range to [-1,1], which limits the impact of individual bit-flips. Therefore, in BitNet, flipping a single bit merely toggles between two extreme values (e.g., -1 and +1), necessitating a substantially more significant number of bit-flips to degrade performance meaningfully.

### 5.3.3 Comparison with Conventional BFA

We evaluated LLMs under the conventional BFA framework Rakin et al. (2019), and observed no noticeable degradation in accuracy even after introducing up to 1,000 bit-flips. These findings suggest that traditional BFA techniques are largely ineffective against large-scale models. In contrast, our proposed AttentionBreaker method induces a substantial drop in model accuracy, as illustrated in Table 4. Unlike conventional BFA approaches, our method employs a gradient-driven strategy for identifying vulnerable layers, selectively targets critical weight subsets, and utilizes GenBFA, a bit selection optimization algorithm. This principled approach allows AttentionBreaker to conduct significantly more effective and scalable attacks on large models.

Table 6: Targeted attack by AttentionBreaker.

| Model Variant | Overall Accuracy (%) | Class A→B Misclassification rate (%) |
|---|---|---|
| Original model | 50.6 | 14.8 |
| Targeted attack by AttentionBreaker (109 bit-flips) | 36.2 | 70.4 |

We further present a comparative analysis of the computational overhead associated with the conventional BFA approach Rakin et al. (2019) and our proposed AttentionBreaker, conducted on the LLaMA3-8B-Instruct INT8 model using four NVIDIA A100 GPUs. As shown in the Table 5, AttentionBreaker achieves approximately $10\times$ faster identification of 1,000 critical bit-flip locations compared to BFA. The inefficiency of traditional BFA arises primarily from its one-bit-per-iteration strategy and the substantial computational cost incurred by repeated gradient recalculations and loss calculations after each bit-flip in each layer. Additionally, some informed decisions can be taken to make the AttentionBreaker even faster, such as only considering the top-1 layer for optimization, reducing the loss threshold etc.

### 5.3.4 Feasibility of Targeted Attacks by AttentionBreaker

While our approach is designed for untargeted attacks as discussed in Section 4.1, it can be adapted for targeted settings by modifying the objective and flipping bits critical to a specific class.

Towards this, let us mount a 1-to-1 targeted misclassification attack on the MMLU benchmark "Astronomy" task using the LLaMA-3.2-1B-Instruct-W8 model. The goal was to cause inputs originally belonging to Class A to be intentionally misclassified as Class B. The experiment involved four classes (A, B, C, D), with model accuracy and targeted misclassification rates measured before and after the attack. In this setting, the goal is not only to degrade overall performance but to enforce targeted misclassification while preserving the accuracy of non-targeted classes. To capture this, we define two complementary loss components.

The attack loss $\mathcal{L}_{\text{attack}}$, which enforces misclassification from class $A$ to $B$, is given by:

$$\mathcal{L}_{\text{attack}} = -\frac{1}{|\mathcal{D}_A|} \sum_{(x,y=A)} \log p_\theta(B \mid x) \tag{12}$$

where $\mathcal{D}_A$ denotes the set of samples with ground-truth label $A$.

The correctness loss $\mathcal{L}_{\text{correct}}$, which preserves classification accuracy for the other classes $\{B, C, D\}$, is defined as:

$$\mathcal{L}_{\text{correct}} = -\frac{1}{|\mathcal{D}_{\neg A}|} \sum_{(x,y\in\{B,C,D\})} \log p_\theta(y \mid x), \tag{13}$$

where $\mathcal{D}_{\neg A}$ is the set of samples not belonging to class $A$.

The total loss $\mathcal{L}_{\text{total}}$ combines the two with a balancing hyperparameter $\lambda$:

$$\mathcal{L}_{\text{total}} = \mathcal{L}_{\text{attack}} + \lambda \mathcal{L}_{\text{correct}}. \tag{14}$$

Minimizing $\mathcal{L}_{\text{total}}$ enforces that inputs from class $A$ are consistently misclassified as class $B$, while predictions for the remaining classes remain accurate. This design ensures that the targeted attack is both focused and stealthy.

Table 6 presents the model's performance in two conditions: the original model (unmodified) and the model after applying AttentionBreaker with only 109 bit-flips to accomplish the targeted attack. As shown in the Table 6, the A→B misclassification rate increased from 14.8% (original model) to 70.4% (when attacked using AttentionBreaker), a 375% relative increase, demonstrating that our method successfully induces targeted

Table 7: Scalability of AttentionBreaker to Larger Models.

| Model | Model type | Precision | # Parameters | # Bit-Flips | WikiText-2 Perplexity Before/After attack | MMLU Accuracy (%) Before/After attack |
|---|---|---|---|---|---|---|
| MiniMixtral-v0.2 | LLM (MoE) | BFloat16 | 12.9B | 2 | 20.78 / $2.1 \times 10^4$ | 58.55 / 0 |
| LLaMA 3.3 70B Instruct | LLM | NF4 (4-bit) | 70.6B | 1 | 9.53 / $1.2 \times 10^4$ | 86.0 / 0 |

behavior. This increase comes with a modest drop in overall accuracy (from 50.6% to 36.2%). These results demonstrate that AttentionBreaker is effective in mounting successful attacks in targeted scenarios.

### 5.3.5 Performance-cost tradeoff

The comparison in the paper in Table 2 evaluates the number of bit-flips needed in the baseline to match AttentionBreaker's performance drop, ensuring a meaningful efficiency assessment. We also provide an alternative comparison by first applying the baseline attack and then matching its degradation with our method. The results in Table 9 shows that AttentionBreaker is significantly more efficient, requiring up to $1.95 \times 10^5$ fewer flip-bits over the baseline method. Our attack framework also provides precise control over the trade-off between number of bit-flips and model loss by explicitly formulating a min-max optimization.

### 5.3.6 Scalability of the evolutionary optimization to larger models (e.g., 70B+ parameters)

We evaluated AttentionBreaker on a Mixture-of-Experts (MoE) architecture. In the MiniMixtral-v0.2 model (12.9B parameters, BFloat16 precision), only two carefully chosen bit-flips were sufficient to cause catastrophic collapse, reducing MMLU accuracy from 58.55% to 0 and increasing WikiText-2 perplexity from 20.78 to $2.1 \times 10^4$. This demonstrates that evolutionary optimization remains effective even in more complex architectures such as MoE.

We fully agree that scaling experiments to 70B+ parameter models is critical. However, due to resource limitations in an academic environment ($4 \times$ NVIDIA A100 GPUs with 320 GB total memory), we cannot feasibly compute full gradient-based sensitivity for models of this scale, as training or gradient evaluation would exceed our compute budget. To still address this point constructively, we performed an alternative gradient-free magnitude-based sensitivity analysis on the LLaMA 3.3 70B Instruct model in 4-bit NF4 quantization. Using magnitude-based sensitivity scoring (as discussed in Section A.5.2 of the paper), we collected the initial subset of critical weights of size 23488. Following this we employed our GenBFA evolutionary optimization to reduce the most critical subset of bits and found that flipping only a **single** critical bit was sufficient to reduce MMLU accuracy from 86% to 0 and increasing WikiText-2 perplexity from 9.53 to $1.2 \times 10^4$. This result demonstrates that AttentionBreaker scales to very large models and maintains high attack efficacy.

These results demonstrate that AttentionBreaker's evolutionary optimization is not only effective for models up to 13B parameters, including MoE architectures, but also remains practical for large-scale models beyond 70B parameters.

### 5.3.7 Attack validation under noisy bit-flip conditions

There is an importance of evaluating the attack in real-world hardware constraints and noisy bit-flip conditions. In practice, bit-flip injections may be imperfect due to factors such as ECC, and other fault-tolerance mechanisms in modern accelerators. To evaluate this, we performed a preliminary noisy bit-flip experiment after identifying the critical bits targeted by AttentionBreaker. Each intended flip was applied with probability p (0.8–1.0), while unintended flips within the same parameter tensor occurred with probability q (0.0–0.3). The choice of p and q values are primarily system dependent and p near 1 and a non-zero q is an acceptable assumption. Each configuration was executed 100 times and averaged. The results, presented in model loss/model accuracy format in the table, indicate that even under high noise conditions, the attack remains effective. Specifically, over 82% of cases resulted in full accuracy collapse (attack loss > 7 on the LLaMA3-8B-Instruct W8 model, MMLU benchmark), and all cases exhibited a relative accuracy drop exceeding 50%. These findings demonstrate that AttentionBreaker maintains strong efficacy under realistic noisy bit-flip conditions.

Table 8: Effect of varying flip probabilities for targeted and untargeted bits on model performance.

| Probability of flipping the targeted bit (p) → | 0.8 | 0.85 | 0.9 | 0.95 | 1.0 |
|---|---|---|---|---|---|
| Probability of flipping a bit not targeted (q) ↓ | MMLU Loss / Accuracy (in %) | | | | |
| 0.0 | 7.38 / 0 | 9.51 / 0 | 9.9 / 0 | 11.4 / 0 | 13.1 / 0 |
| 0.05 | 7.11 / 0.3 | 7.48 / 0.2 | 9.0 / 0 | 10.8 / 0 | 12.7 / 0 |
| 0.1 | 8.15 / 0 | 8.8 / 0 | 8.1 / 0 | 10.6 / 0 | 11.8 / 0 |
| 0.15 | 7.23 / 0 | 8.9 / 0 | 8.5 / 0 | 11.1 / 0 | 11.4 / 0 |
| 0.2 | 5.33 / 21.4 | 5.8 / 14.8 | 8.4 / 0 | 10.1 / 0 | 10.2 / 0 |
| 0.25 | 6.0 / 15.3 | 7.1 / 2.2 | 7.9 / 0 | 8.6 / 0 | 9.9 / 0 |
| 0.3 | 3.83 / 21.3 | 6.41 / 5.3 | 6.59 / 1.2 | 8.13 / 0 | 9.62 / 0 |

Table 9: Performance-cost tradeoff on baseline Nazari et al. (2024) and the AttentionBreaker.

| Model | Relative accuracy drop (%) | # Bit-Flips in Existing Work Nazari et al. (2024) | # Bit-Flips in AttentionBreaker |
|---|---|---|---|
| LLaMA3-8B-Instruct-W8 | 10 | 5872 | 3 |
| | 25 | 13204 | 3 |
| | 50 | 57235 | 3 |
| | 100 | 587202 | 3 |

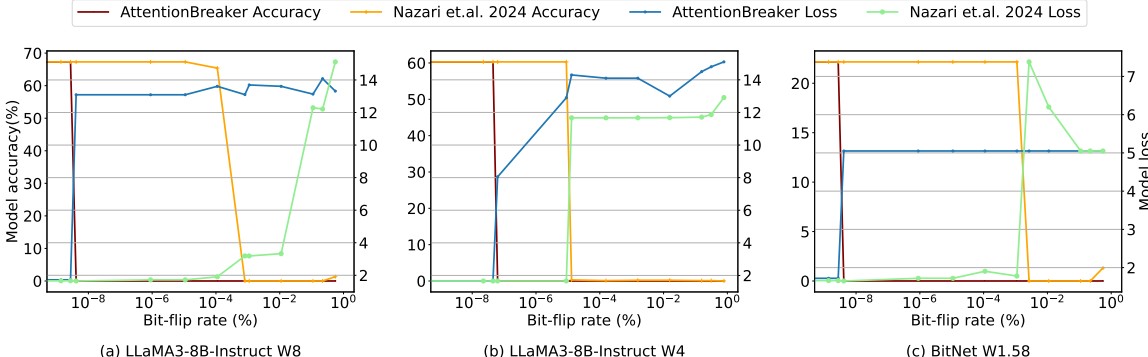

(a) LLaMA3-8B-Instruct W8     (b) LLaMA3-8B-Instruct W4     (c) BitNet W1.58

Figure 3: Comparison of BFA attack using Nazari et al. (2024) and AttentionBreaker on LLaMA3-8B-Instruct W8, W4, and W1.58 models. Metrics are calculated on the "Astronomy" task of the MMLU benchmark.

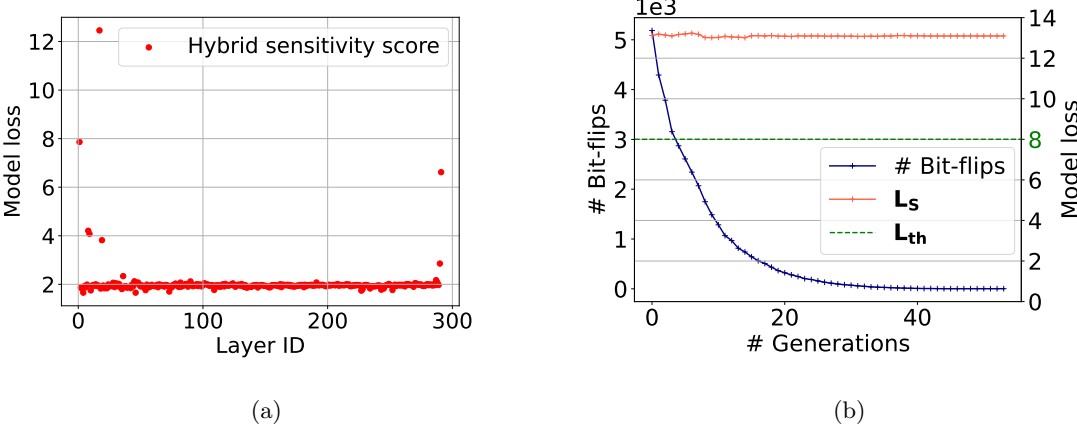

(a)          (b)

Figure 4: (a) Layer sensitivity analysis, (b) evolutionary optimization in LLaMA3-8B-Instruct W8 model.

Table 10: Effect of AttentionBreaker on FaR-based defense.

| Model | Dataset | Original Accuracy (%) | # of Bit-flips | Post-attack accuracy (%) | |
| --- | --- | --- | --- | --- | --- |
| | | | | With FaR | Without FaR |
| ViT-Base | CIFAR-100 | 91.55 | 224 | 3.4 | 4.2 |
| | ImageNet1k | 80.1 | 268 | 10.9 | 10.2 |

## 5.4 Analysis of intermediate steps in AttentionBreaker

### 5.4.1 Assessing Layer Sensitivity and Ranking

In this section, we analyze the impact of the layer sensitivity analysis, described in Section 4.2.1. Figure 4a shows the vulnerability profiles of model layers by plotting model loss against layer IDs, where each layer experiences bit-flip perturbations. The figure reveals increased sensitivity in the initial and final layers for the LLaMA3-8B-Instruct W8 model. Further analysis of other models is in the appendix (Section A.3.1). After the sensitivity analysis, the layers are sorted, and the top-1 layer is chosen for optimization, reducing the search space. For LLaMA3-8B-Instruct W8, selecting the most sensitive layer reduces the solution space from $\approx 8 \times 10^9$ to $\approx 5.8 \times 10^7$ parameters ($\approx 138\times$).

### 5.4.2 Obtaining the Most Critical Weights

In this section, we analyze the impact of the proposed GenBFA evolutionary algorithm, detailed in Section 4.2.2. Figure 4b illustrates the reduction in the weight set while preserving a high model loss for the LLaMA3-8B-Instruct W8 model. As depicted in the figure, the model loss (red line) remains above a predefined threshold (green line), while the number of required bit-flips (blue line) is iteratively minimized. The initial subset size is reduced by a factor of up to $2 \times 10^3$, identifying only **3** critical weights—sufficient to execute the AttentionBreaker attack with effectiveness comparable to perturbing the original set of 5872 weights. A similar trend is observed for Phi-3-mini-128k W8, LLaMA3-8B-Instruct W4, and BitNet W1.58 models (detailed in the appendix – Section A.3.2), highlighting the efficacy of GenBFA.

## 6 Efficiency against Existing Defenses

Existing research proposed a Forget and Rewire (FaR)-inspired defense for protecting against BFAs Nazari et al. (2024). In this section, we evaluate the prowess of AttentionBreaker against FaR. Table 10 demonstrates the effects of AttentionBreaker on a ViT-Base Dosovitskiy et al. (2021) model in the presence of FaR (Nazari et al. (2024) used similar ViT models in their work). Our results reveal that, despite the FaR defense, AttentionBreaker was successful in plummeting the accuracy of ImageNet1k dataset from 80.7% to 10.9% after just 268 ($\approx 3 \times 10^{-6}$%) bit-flips. Thus, it can be concluded that AttentionBreaker is a potent attack that can subvert state-of-the-art defenses deployed for protecting transformer-based models against BFA.

We also have identified SOTA defenses against bit-flip attacks on deep neural networks (DNNs), such as Aegis Wang et al. (2023) and low bit-width quantization based defenses He et al. (2020). Aegis employs a multi-exit mechanism that incorporates multiple internal classifiers, with one randomly selected for output generation in an effort to mitigate attacks Wang et al. (2023). However, this approach is ineffective if an adversary targets early layers, leading to erroneous activations across all classifiers. As demonstrated in our study, AttentionBreaker detects initial layers as vulnerable to bit-flips (refer to Section 6.3.1) and mounts successful attacks, rendering Aegis ineffective. Additionally, deploying early-exit mechanisms in LLMs incurs significant computational overhead, limiting their practicality. Further, having such models requires the exit layers to be trained, which is often infeasible in resource-limited client setups. Low bit-width quantization reduces precision to mitigate bit-flip impacts He et al. (2020). However, AttentionBreaker strategically identifies and manipulates critical bits even in quantized models, inducing severe degradation. For example, in LLaMA3-8B-Instruct (NF4, 4-bit), AttentionBreaker reduces accuracy from 60.3% to 0% with just 28 bit-flips, demonstrating its effectiveness against quantization-based defenses.

# 7 Discussion on Possible Defenses

In response to the attacks exemplified by the proposed AttentionBreaker, we recommend an adaptive defense mechanism designed to disrupt the foundational strategies of the attack. This defense mechanism draws inspiration from XOR-cipher logic locking Kamali et al. (2022) and preemptively identifies a critical set of vulnerable model parameters. A subset of these parameters is then encrypted using an XOR-cipher. With the correct key set, the model retains performance during inference; an incorrect key results in significant performance loss. An attacker unaware of the key faces a model with reduced baseline accuracy, making further attacks impractical.

To experimentally validate the proposed XOR-cipher/logic-locking–based defense, we apply XOR-locking to the vulnerable bits identified by AttentionBreaker, ensuring that only the correct key restores baseline accuracy, while any incorrect key leaves the model in a severely degraded state, thereby deterring further attacks. The security scales exponentially with key length; for example, locking 100 bits with a 100-bit key yields $2^{100}$ possible combinations, making brute-force infeasible. To harden protection when the critical subset is small (e.g., 3 bits), we lock a larger superset. In one experiment with LLaMA3-8B-W8, locking 100 bits restored 67.3% MMLU accuracy with the correct key, but incorrect keys reduced accuracy to an average of 26.7%, often causing complete collapse. Furthermore, once the model is degraded by an incorrect key, any subsequent attack with AttentionBreaker faces low incentive and high inefficiency: the optimization process aims to degrade performance that is already poor. Thus, without the correct key, the model is functionally unusable, and follow-up adversarial efforts are unattractive and largely ineffective.

# 8 Ethical Implications

## 8.1 Ethical Considerations and Potential Misuse

The AttentionBreaker framework, while intended to expose critical vulnerabilities in LLMs, presents clear dual-use risks. Its ability to induce total model failure with minimal bit-flips, even bypassing state-of-the-art defenses, could be maliciously exploited, particularly in cloud or shared hardware environments. Such misuse threatens the reliability of LLMs in safety-critical domains like healthcare, finance, and security. Given this potential, responsible disclosure is imperative. The release of related tools should be tightly controlled, with coordination among AI developers, hardware vendors, and cloud providers. Ethical oversight akin to review boards in other high-risk research domains is essential to prevent abuse. In conclusion, AttentionBreaker highlights the fragility of current AI systems and underscores the urgent need to treat adversarial robustness as a core design objective, rather than a post-hoc safeguard.

## 8.2 Dual-Use Risks (e.g., Attacks via Shared Hardware)

While our work focuses on demonstrating the vulnerability of LLMs to bit-flip attacks, we acknowledge the dual-use potential of such techniques. This is particularly crucial in scenarios where shared or untrusted hardware is used, as in multi-tenant MLaaS deployments. In such environments, an adversary could exploit hardware faults (e.g., RowHammer Mutlu & Kim (2019), RowPress Luo et al. (2023)) to induce critical bit-flips without direct access to model weights. These risks are amplified in shared cloud hardware where multiple users' workloads may co-locate on the same physical accelerator, providing an opportunity for malicious tenants to target co-resident models. By identifying highly vulnerable layers and bits, AttentionBreaker could be repurposed by an adversary to design efficient hardware-level fault injection campaigns, underscoring the need for proactive hardware and deployment-level safeguards.

## 8.3 Regulatory and Policy Implications

The findings of this work suggest that hardware-level protections should be integral to MLaaS security frameworks. Potential countermeasures include memory isolation enhancements, logic locking, and physically unclonable functions (PUFs) to authenticate accelerator hardware. From a governance perspective, regulatory bodies could mandate model providers to perform red-team testing, deploy certified secure hardware for hosting sensitive models, and document bit-flip resilience in compliance reports. Emerging AI

policy proposals, including compute governance and hardware licensing regimes, also highlight the growing recognition that model security cannot be separated from hardware security. Incorporating such measures can help mitigate the real-world risk of bit-flip exploitation in MLaaS settings.

## 9 Conclusion

In this work, we introduced a novel adversarial BFA framework, AttentionBreaker, for transformer-based models such as LLMs. For the first time, AttentionBreaker applies a novel sensitivity analysis along with evolutionary optimization to guide adversarial attacks on LLMs, leading to degraded performance. Specifically, perturbing merely **3**-bits $4.129 \times 10^{-9}$% of the parameters in LLaMA3-8B-Instruct W8 results in MMLU scores dropping from 67.3% to 0% and Wikitext perplexity increased sharply from 12.6 to $4.72 \times 10^5$. Furthermore, our experimental results demonstrate that AttentionBreaker can subvert state-of-the-art defenses for protecting transformer-based models against BFA. These findings underscore the effectiveness of AttentionBreaker in exposing the vulnerability of LLMs to such adversarial interventions.

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

# A Appendix

## A.1 Background

In this section, we elaborate on the types of model attacks based on the level of knowledge the attacker can extract from the victim model.

### A.1.1 Model Attacks

**Full Knowledge Attacks** In shared cloud environments, attacks leveraging complete knowledge represent the gravest security threat to deployed large language models (LLMs). This scenario assumes an adversary with extensive knowledge of both the target model and deployed defenses, thus allowing for optimized attack strategies. Attackers can utilize advanced side-channel techniques to compromise hardware vulnerabilities and exfiltrate sensitive model parameters, encompassing proprietary weights and confidential user queries. One of the most potent techniques available to attackers in this context is the Cache Side-Channel Attack, specifically targeting the Last-Level Cache (LLC). Yarom & Falkner (2014) and Kayaalp et al. (2016) have shown that these attacks leverage shared hardware resources to deduce memory access patterns, thereby exposing sensitive data within the cache. Despite the severity of full-knowledge attacks, they remain technically challenging due to limited data extraction rates and the high complexity of reconstructing deep neural networks from indirect leakage. However, even the partial extraction of key model components, including embedding layers and attention heads, poses a significant risk, potentially enabling adversaries to replicate model functionality or generate targeted adversarial perturbations.

**White-box Attacks:** White-box attacks pose a serious threat to large language models (LLMs), leveraging attackers' knowledge of model architecture and training data Madry (2017); Rakin et al. (2019). While attackers cannot directly access memory, they can estimate vulnerable bits to execute Rowhammer attacks and exploit hardware vulnerabilities Mutlu & Kim (2019).

Two key factors contribute to this risk. First, many commercial models and datasets are open-source, and users frequently download pre-trained models for transfer learning. These models inherit vulnerabilities, giving attackers a foundation for exploitation. Second, adversaries can use hardware monitoring tools to analyze memory access patterns Yan et al. (2020), allowing them to infer model architecture and train a local copy with similar weaknesses. Unlike full-knowledge attacks, white-box attacks are practical and accessible, making them a baseline threat model. Mitigating these risks requires robust memory protection, hardware-level defenses, and adversarial training to prevent model extraction and fault injection.

**Gray-box Attacks:** In gray-box scenarios, attackers possess partial knowledge of the target model, such as its architecture or certain parameters, but lack complete access. This limited insight enables them to craft more efficient attacks compared to black-box methods Xiang et al. (2020); Zhang & Sabuncu (2018). Attackers analyze model outputs and leverage statistical or adversarial techniques to infer model properties and vulnerabilities or extract knowledge. These attacks enable techniques such as model extraction (where attackers approximate the target model's behavior) and prompt inference (where sensitive information is retrieved based on response patterns).

Since attackers lack direct access to model internals, they rely on query efficiency to reduce costs while maximizing the extraction of decision boundaries or key model parameters. This attack type is particularly concerning in API-based LLM deployments, where adversaries can probe responses and fine-tune adversarial inputs without needing access to underlying weights or architecture.

**Black-box Attacks:** In black-box scenarios, attackers have no direct access to the target model's architecture or parameters Papernot et al. (2017); Mahmood et al. (2021). They rely solely on the model's outputs to infer vulnerabilities and craft adversarial examples. A notable approach involves using gradient estimation techniques to generate adversarial inputs, effectively deceiving models without internal knowledge. Another study highlighted the practicality of black-box attacks by crafting adversarial examples that mislead machine-learning models, even in the absence of model specifics.

**Real-world Execution Feasibility:** In this work, we use the white-box attack setting for identifying the most critical bits in the model. Even though a white-box type bit flip attack assumes direct access to model weights, an attacker can still indirectly induce such faults in a deployed model by targeting the hardware that holds the model in memory. In real-world scenarios, attackers might exploit hardware vulnerabilities—such as the Rowhammer effect—to cause bit flips in DRAM. For example, if an attacker manages to run code on the same physical server as the deployed model (as might be possible in a multi-tenant cloud environment), they can use fault injection techniques to flip a few critical bits in memory identified by techniques such as Memory Access Pattern Analysis Yan et al. (2020). This doesn't require direct API access to the model's weights; rather, it leverages a vulnerability in the underlying hardware to corrupt the model's parameters at runtime.

### A.1.2 Transformer Models

The Transformer architecture is foundational for LLMs and Vision Transformers, driving advancements in NLP and image classification Vaswani (2017); Radford (2018); Chang et al. (2024); Xu et al. (2024). It comprises two main components: the encoder to convert input data into a structured intermediate representation for improved feature understanding and the decoder to generate output sequences from this encoded informationZhang et al. (2023a). There are three task-specific configurations: Encoder-only models for input-driven tasks (e.g., text classification, named entity recognition), Decoder-only models for generative tasks, and Encoder-decoder models for tasks requiring input-dependent generation (e.g., machine translation, summarization).

A critical innovation in the Transformer is the attention mechanism, particularly multi-head attention, which prioritizes relevant input elements Zhang et al. (2023a). Attention weights are computed to emphasize informative tokens, while masks can exclude specific tokens, such as padding. In each multi-head attention layer, the model derives three parameters for each input token: **Query** ($Q$), **Key** ($K$), and **Value** ($V$), calculated through linear transformations of input vectors. Attention scores assess token importance across the sequence. Each head processes distinct $Q$, $K$, and $V$ subsets, facilitating parallel analysis of diverse token relationships. The individual attention outputs are concatenated and linearly combined to produce the final attention output, enhancing the model's ability to capture complex dependencies and improve task performance in both NLP and vision domains.

### A.1.3 Bit-flip Attack (BFA)

**Bit-flip Vulnerabilities:** The robustness, security, and safety of modern computing systems are intrinsically linked to memory isolation, which is enforced through both software and hardware mechanisms Frassetto et al. (2018). However, due to read-disturbance phenomena, contemporary DRAM chips remain susceptible to memory isolation breaches Kim et al. (2014). RowHammer Mutlu & Kim (2019) is a well-documented example where repeated DRAM row (hammering) access induces bitflips in adjacent rows due to electrical interference. SledgeHammer enhances the effectiveness of RowHammer by combining it with bank-level parallelism and hammering multiple memory banks simultaneouslyKang et al. (2024). A recent and significant advancement in read-disturbance attacks is RowPress Luo et al. (2023), specifically targeting DDR4 systems with RowHammer protection mechanisms. RowPress demonstrates that prolonged access to a single DRAM row can disrupt physically adjacent rows, causing bitflips. This technique effectively circumvents RowHammer defenses, requiring substantially fewer row activations to induce faults and underscoring DRAM's vulnerability.

**Existing Bit-flip Attacks:** Existing research has explored various fault injection methods in neural networks, targeting biases, weights, and activation functions to induce targeted and untargeted misclassifications. While these techniques were primarily designed for full-precision DNNs, many edge implementations operate in quantized precision, making them inherently more resilient to parameter variations. To overcome this, efficient algorithms have been developed to identify the most vulnerable bits in quantized DNNs. Rakin et al. Rakin et al. (2019) exploit RowHammer to expose BFA vulnerabilities in DNNs, using Progressive Bit Search (PBS) to iteratively flip critical bits and degrade accuracy. TBFA Rakin et al. (2021) extends this by targeting specific misclassifications. DeepHammer Yao et al. (2020) exemplifies advanced BFAs by

employing system-level flip-aware PBS that enables memory-efficient rowhammering and precise flipping of targeted bits.

However, BFAs have been predominantly studied in the context of DNNs and remain largely unexplored for Transformer-based architectures such as LLMs due to their substantially larger parameter space and structural differences. Applying conventional BFA techniques to LLMs is highly challenging, a difficulty further exacerbated by the growing adoption of quantized and low-precision models. Towards filling this research gap, we for the first time propose a novel, efficient BFA framework, AttentionBreaker, to methodically exploit the bit-flip vulnerabilities of LLMs in an efficient manner. While our approach is designed for untargeted attacks, it can be adapted for targeted settings by modifying the objective and flipping bits critical to a specific class. We focus on untargeted attacks as earlier research indicates that they are more challenging to defend Özdenizci & Legenstein (2022). Studies further suggest that degrading overall performance through untargeted attacks lacks distinct attack signatures unlike targeted attacks, rendering conventional defenses ineffective Qian et al. (2023); Nazari et al. (2024). Therefore, AttentionBreaker strategically employs an untargeted attack to maximize disruption while evading traditional defense strategies.

**Quantization precision impact on bit-flip attacks:** Quantization precision plays a crucial role in determining the vulnerability of machine learning models to bit-flip attacks. It affects both the number of bit-flips required and the extent to which these flips can disrupt model performance. The impact of bit-flips varies across different numerical formats due to their underlying representation and sensitivity to perturbations. The value of an integer stored in an n-bit signed representation (e.g., INT8, INT4, INT2) is given by:

$$X = (-1)^{b_{n-1}} \sum_{k=0}^{n-2} b_k \cdot 2^k \tag{15}$$

Where $b_k$ represents the bit at position $k$. Flipping a bit at position $k$ changes $X$ by:

$$\Delta X = (-1)^{b_{n-1}} \cdot 2^k \tag{16}$$

On the other hand, floating-point representations, such as FP16 and FP32, follow the IEEE 754 standard, where a number is stored as:

$$X = (-1)^s 2^{E-E_{\text{Bias}}}(1 + M) \tag{17}$$

where $s$ is the sign bit, $E$ is the exponent with bias $E_{\text{Bias}}$, and $M$ is the normalized mantissa. Now, flipping the sign bit changes the sign of the value. Flipping a bit in the exponent can change the value by a factor of $2^k$, leading to drastic value shifts (e.g., a bit-flip in the 7th exponent bit transforms an FP32 number 1.0 to infinity). Flipping a bit in the mantissa introduces a small perturbation, where the relative change is approximately $2^{-23}X$, which is much smaller and therefore requires multiple bit-flips in the mantissa to achieve significant impact.

For example, in INT8, flipping the most significant bit (MSB) of the value (0b01000000) changes from 64 to -64. This demonstrates that MSB flips can significantly alter values, but the effect is constrained by the limited range of [-128, 127]. As the precision further declines to INT4 and INT2, the range correspondingly diminishes, resulting in ranges of [-8, 7] and [-1, 1] respectively. Consequently, a single bit-flip causes a smaller error, necessitating a larger number of bit-flips to produce a significant effect.

In a Float32 parameter, flipping the most significant bit can change a value from a large positive number to an equally large negative number (e.g., $3.4 \times 10^{38}$ to $-3.4 \times 10^{38}$). This drastic shift reflects the role of the sign bit in IEEE 754 representation. Bit-flips in the exponent or mantissa can further lead to outcomes ranging from numerical overflow to subtle degradation. This vulnerability extends to reduced-precision formats. In Float16 (FP16), with a smaller dynamic range and 10-bit mantissa, sign-bit flips produce equivalent sign inversion (e.g., $6.5 \times 10^4$ to $-6.5 \times 10^4$), while exponent or mantissa flips yield more pronounced errors due to limited precision. Similarly, bfloat16 (BF16), which retains Float32's 8-bit exponent but reduces the mantissa to 7 bits, exhibits analogous sign sensitivity with coarser representational granularity. Table 11 depicts the extreme criticality of full-precision models. For instance, in LLaVA1.6-Mistral-7B FP16, a

**single bit-flip** via AttentionBreaker suffices to collapse model accuracy entirely, whereas the W4 (NF4) variant requires 53 flips to achieve a comparable effect. These findings underscore that, despite appearing more robust, lower-precision models remain extremely vulnerable to bit-flip attacks. Therefore, we focus on targeting lower precision models in our experiments.

| Model | Model Type | Precision | # Parameters | AttentionBreaker (# Bit-Flips) | Benchmarks (Accuracy %) Before / After Attack | | | |
|---|---|---|---|---|---|---|---|---|
| | | | | | WikiText-2 | MMLU | VQAv2 | TextVQA |
| LLaMa-3 | LLM | BFloat16 | 8.03B | 2 | 10.24 / 3.8e4 | 66.3 / 0 | NA | NA |
| LLaVa Mistral 1.6 | VLM | BFloat16 | 7.57B | 1 | NA | NA | 77.2 / 0 | 63.6 / 0 |
| Phi-3-mini-128k | LLM | BFloat16 | 3.82B | 3 | 9.45 / 3.6e4 | 68.1 / 0 | NA | NA |
| LLaMa-3.2 | LLM | BFloat16 | 1.24B | 2 | 20.78 / 2.1e4 | 49.3 / 0 | NA | NA |
| Qwen3-0.6B-Base | LLM | BFloat16 | 596M | 1 | 18.23 / 3.2e4 | 49.6 / 0 | NA | NA |
| Deepseek-7B | LLM | BFloat16 | 7B | 4 | 9.92/4.2e4 | 41.8/0 | NA | NA |
| MiniMixtral-v0.2 | LLM | BFloat16 | 12.9B | 2 | 20.78/2.1e4 | 58.55/0 | NA | NA |

Table 11: Effectiveness of AttentionBreaker across multiple full-precision models and benchmarks.

## A.2 Ablation Study

This section presents an ablation study to systematically evaluate and decide on specific components in our proposed method, such as $\alpha$, top-$n$, or which genetic algorithm to use for optimization.

### A.2.1 Sensitivity Ratio Variation

We vary $\alpha$ between $|\nabla \mathbf{W}|$ and $|\mathbf{W}|$ in the min-max normalized importance score calculation as described in Section 4.2.1. Under different values of $\alpha$, the model loss at different bit-flip rates is depicted in Figure 5a. As observed, an optimal value of $\alpha = 0.5$ achieves the best results, closely followed by $\alpha = 0$ and $\alpha = 0.75$. *Consequently, $\alpha = 0.5$ is selected for subsequent experiments.*

### A.2.2 All vs top-$n$ Layer Sensitivity Analysis

In this experiment, we investigate the impact of varying the selection of top-$n$ layers on model performance by comparing the effects of sub-sampling from the top-$n$ layers versus sub-sampling from all layers. With a fixed sensitivity parameter of $\alpha = 0.5$, we increment $n$ from 1 to 5. The results demonstrate that sub-sampling from the top-$n$ layers leads to a higher loss than sub-sampling from all layers, as illustrated in

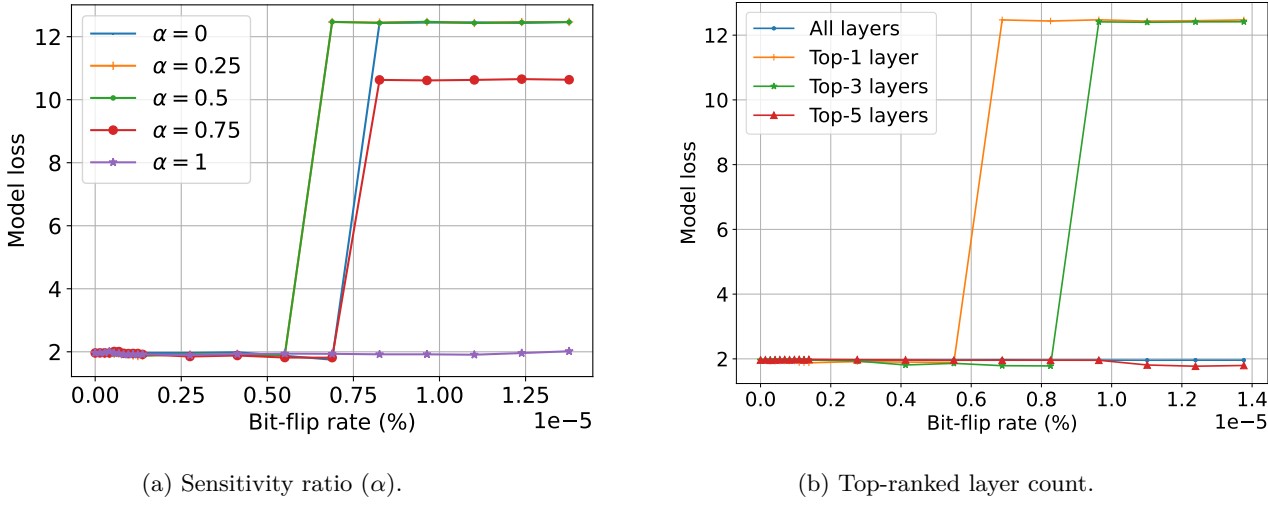

(a) Sensitivity ratio ($\alpha$).      (b) Top-ranked layer count.

Figure 5: LLaMA3-8B-Instruct W8 ablation study on Sensitivity Ratio Variation and top-$n$ layer count. Analysis was performed using the 'Astronomy' task of the MMLU benchmark.

Figure 5b. Among the top-$n$ layers, selecting the top-1 layers yields the best performance, followed by top-3 and top-5. The weight subset selection and optimization based on top-$n$ layers provide superior results, as the constrained weight subspace facilitates faster convergence. *Consequently, we fix $n = 1$ for all subsequent experiments.*

### A.2.3 Genetic Algorithms Variation

We extensively evaluate the performance of our proposed evolutionary optimization technique against established algorithms, including the Simple Genetic Algorithm (SGA) Vose (1999), Particle Swarm Optimization (PSO) Eberhart & Kennedy (1995), Non-Dominated Sorting Genetic Algorithm (NSGA) Srinivas & Deb (1994), and NSGA-II Deb et al. (2000); Lambora et al. (2019). It is essential to note that these baseline methods employed are established benchmarks in recent research Kundu et al. (2024b); Zhang et al. (2023b); Wang et al. (2024). Our problem is inherently multi-objective, as we aim to maximize model loss and minimize the number of bit-flips. These two objectives make NSGA-based methods particularly relevant for comparison. We reformulated our approach into a unified loss function to ensure a fair evaluation. This transformation allows for a direct comparison between our method and NSGA-based approaches while preserving the underlying optimization goals. To ensure comparability, we standardize parameters across all algorithms, setting the population size to 100, mutation rate to 0.1, and crossover rate to 0.9. Additionally, we apply modifications in population initialization, crossover, and mutation strategies as in our custom implementation. Results in Figure 6a demonstrate that our modified genetic algorithm achieves superior convergence speed and accuracy, owing to optimizations tailored to problem-specific requirements. *Consequently, this optimized genetic algorithm is employed in all subsequent experiments.* Figure 6b illustrates the optimization of the weight set within AttentionBreaker's genetic implementation. Notably, the required weight count decreases exponentially as the fitness score rises, reflecting rapid improvement. After approximately 80–90 iterations, both metrics plateau, indicating convergence; at this stage, the optimal solution is returned as the output.

### A.3 Intermediate Results

### A.3.1 Layer Sensitivity Analysis

Figure 7 presents the sensitivity distribution using loss across layers for the LLaMA3-8B-Instruct W4, along with BitNet and LLaVA1.6-7B models. For LLaMA3-8B-Instruct W4, the model loss across several layers remains elevated, signifying high sensitivity in Figure 7a. Figure 7b describes the sensitivity of BitNet,

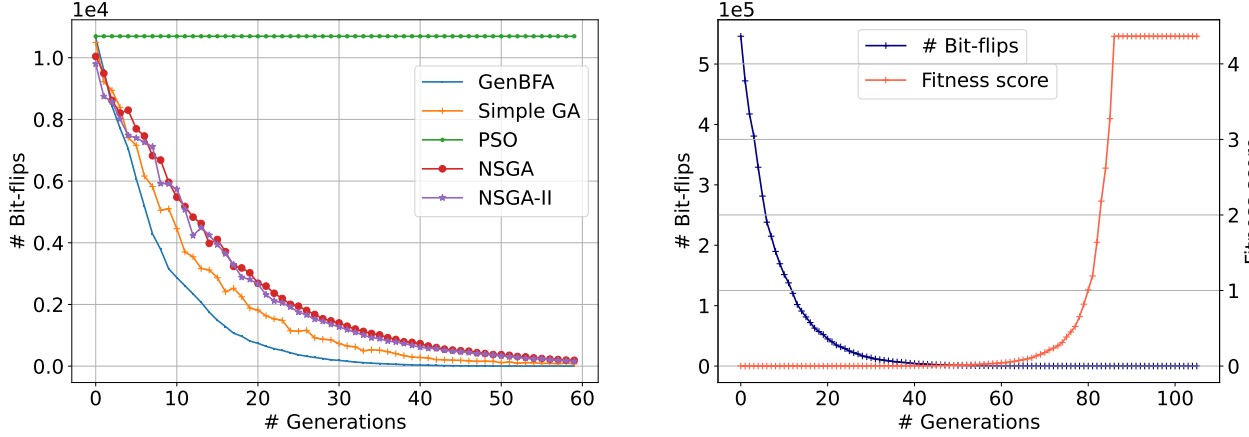

(a) comparison between various genetic algorithms.

(b) number of bit-flips and fitness score optimization.

Figure 6: LLaMA3-8B-Instruct W8 ablation study on genetic optimization to select critical weights. Fitness scores for the genetic algorithms are calculated using the "Astronomy" task of the MMLU benchmark.

which, with its more aggressive quantization, exhibits greater resilience, reflected by fewer layers furnishing high model loss. In the case of LLaVA1.6-7B W8, high sensitivity to parameter perturbation is observed primarily in the language processing component of the model. The vision processing component, preceding the language component, has a less pronounced effect. The most sensitive layers are among the initial layers of the language component, as seen in Figure 7c. This showcases the variation in layer sensitivities in different components in multi-modal models.

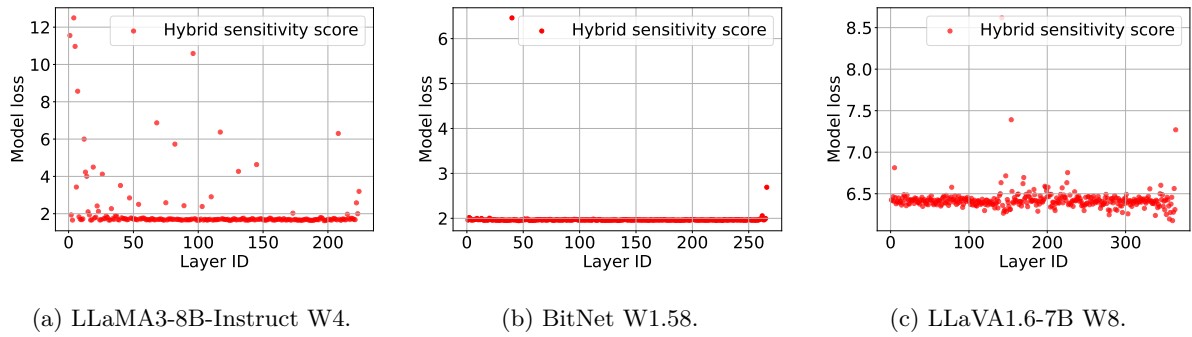

(a) LLaMA3-8B-Instruct W4.  (b) BitNet W1.58.  (c) LLaVA1.6-7B W8.

Figure 7: Layer sensitivity analysis for LLMs and VLMs with varying quantization formats. Sensitivity scores and model loss are calculated using the 'Astronomy' task of the MMLU benchmark.

### A.3.2  OBTAINING THE MOST CRITICAL PARAMETERS

The GenBFA algorithm, detailed in Section 4.2.2, refines the weight subset generated by the previous step. Through iterative refinement, this stage systematically reduces the weight subset, yielding a final set for bit-flip perturbation that is minimal in size yet achieves the same degradation in model performance as the initial set. This iterative optimization process is captured in Figure 8, which shows an exponential reduction in weight subset size while maintaining loss at or higher than the loss threshold.

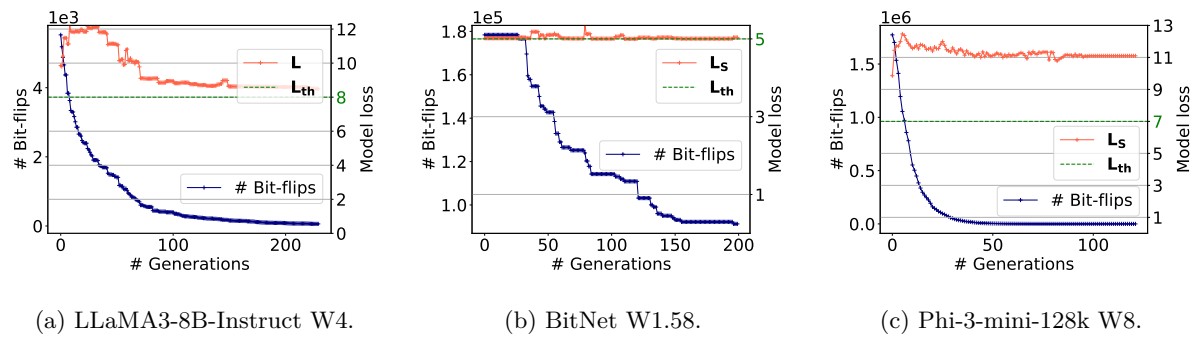

(a) LLaMA3-8B-Instruct W4.  (b) BitNet W1.58.  (c) Phi-3-mini-128k W8.

Figure 8: Genetic optimization for critical parameter identification. Loss is calculated on the 'Astronomy' task on MMLU.

### A.4  Illustrative Attack Efffects of the Model Attack

In this section, we systematically analyze the impact of adversarial bit-flip perturbations on model behavior, as introduced through our proposed framework. We present empirical evaluations across a diverse set of large language model (LLM) tasks, including question answering, code generation, content generation, and translation. The following box illustrations provide representative examples demonstrating the extent of model degradation under BFAs.

### A.4.1 Mathematical question answering

Box 2 presents a comparative analysis of the model's behavior before and after the adversarial attack. The results illustrate a substantial degradation in the model's output quality following the attack. When tested with LLaMA3-8B-Instruct W8 with MMLU under normal operation, the model produces accurate and contextually relevant responses. However, post-attack, the generated output becomes highly erroneous and incoherent with respect to the given query, highlighting the significant impact of the adversarial perturbation on the model's reliability.

### A.4.2 Code generation

Box 3 presents an illustrative example of the model's behavior in a code generation task before and after the adversarial attack. In the pre-attack scenario, the model correctly generates a Python function to compute the Fibonacci sequence, demonstrating its intended functionality. However, following the attack, the model's output becomes completely corrupted, consisting of incoherent and nonsensical characters. This stark contrast highlights the severe impact of the adversarial perturbation, leading to a catastrophic failure in code generation and rendering the model unreliable for practical use.

### A.4.3 Language translation

Box 4 illustrates the impact of the adversarial attack on a translation task. In the pre-attack scenario, the model correctly translates the given English sentence into French, producing the expected output with high fidelity. However, after the attack, the model fails to perform the translation entirely, instead echoing the original English input followed by a sequence of nonsensical characters. This degradation demonstrates the attack's effectiveness in disrupting the model's core functionality, leading to a complete failure in producing a coherent translation.

---

**Box 2: Mathematical question answering**

**System Prompt**: You are a helpful AI assistant. Answer the following question.
**Few-shot (n) Examples**:
**Question 1**
**Expected model output 1**
**Question 2**
**Expected model output 2**
...
**Question # (n+1):** Edgar eats 18 pretzels a day. If his brother eats 1/2 as many, how many does his brother eat in a week?

**Model Output (Pre-Attack)**

**Model Output:**
Edgar eats 18 pretzels a day. His brother eats 1/2 as many, so he eats 18 * 1/2 = 9 pretzels a day. In a week, he eats 9 * 7 = 63 pretzels. The answer is 63.
**Extracted Answer:** 63

**Model Output (Post-Attack)**

**Model Output:**
/iwbfninwdjo[[jj cweijfnpow[c w edfjo [wedj [wed p owed[ wqd
**Extracted Answer:**""

---

**Box 3: Code generation**

**System Prompt**: Write a function to Write a code to calculate fibonacci sequence in python.

Model Output (Pre-Attack)

**Model Output:** If we have to write a code in python to calculate the Fibonacci sequence for the given input of 'n', we can create a function called 'fibonacci' as follows:
def fibonacci(n):
# initializing the first two fibonacci numbers
a, b = 0, 1
# printing the first two numbers
if n > 0:
  print(a, b)

Model Output (Post-Attack)

**Model Output:** ...!!!!!!!!!!!!!! !!!!!!!! !!!!!!! !!!!!!!!!!!!!!!!!!!!!!!!!!!!!!!!!!!!! !!!!!!!!!!!!!!!!!!!!!!!!!!!!!!!!!!!!!!!! !!!!!!!!!!!!!!!!!!!!!!!!!!!!!!!!!!!!!!!

**Box 4: Language translation**

**System Prompt**: Translate from English to French: I have to submit an assignment tomorrow.
**Expected output**: Je dois soumettre un devoir demain.

Model Output (Pre-Attack)

**Model Output:** Je dois soumettre un devoir demain.

Model Output (Post-Attack)

**Model Output:** I have to submit an assignment tomorrow.!!!!!!!!!!!!!!!!!!!!!!!

**Box 5: Content generation**

**System Prompt**: Write an article about "Artificial Intelligence and Ethics."

Model Output (Pre-Attack)

**Model Output:** Artificial intelligence (AI) has been a field of interest and innovation, promising numerous profound impacts on society. However, as AI technology advances,...

Model Output (Post-Attack)

**Model Output:** Write an article about Artificial Intelligence and Ethics:ethesisethesis@ps@ peaceHub@ peace@K@2@K@@K@@2@K[@@2...

### A.4.4 Content generation

Box 5 highlights the adverse effects of the attack on a content generation task. Before the attack, the model generates a coherent and contextually relevant article introduction on "Artificial Intelligence and Ethics", demonstrating its ability to produce structured and meaningful text. However, after the attack, the model's output becomes entirely corrupted, consisting of a mix of nonsensical phrases, symbols, and numerical sequences. This severe degradation underscores the attack's capability to disrupt the model's generative capabilities, rendering it incapable of producing intelligible or contextually appropriate content.

## A.5 Adaptability and Robustness

In this section, we examine the adaptability of AttentionBreaker in constrained environments, along with its robustness and the transferability of its attack effects.

### A.5.1 Gradient-free AttentionBreaker

In this experiment, we investigate the effectiveness of AttentionBreaker in a gradient-free context, where gradient information is inaccessible at all stages. Here, the importance scores are computed solely based on weight magnitudes by setting $\alpha = 0.0$. Our results demonstrate that this magnitude-based approach maintains the attack's efficacy, as it accurately identifies critical weights, enabling a highly impactful attack. We observe that, without gradient information, setting $\alpha = 0$ allows the model to be attacked with only $\mathbf{4.129 \times 10^{-9}}\%$ bit-flips, reducing model accuracy to zero. This degradation matches the attack's effectiveness at $\alpha = 0.5$, as depicted in Figure 3a. Subsequent analysis of the attack vector confirms that identical weights and bits are selected for attack across both settings, underscoring AttentionBreaker's versatility and adaptability. Additionally, Section 5.3.6 demonstrates how gradient-free AttentionBreaker can be highly capable in attacking large models such as LLaMA 3.3 70B Instruct NF4 model with over 70 billion parameters. This robustness is particularly significant as many defenses against BFAs assume that gradient access is essential for an attacker, prompting defenses to obscure or protect gradients. *However, AttentionBreaker's gradient-free capability bypasses these gradient-based defenses entirely by relying on magnitude-based importance, showing its potential to circumvent gradient-based restrictions. This adaptability highlights Attention-Breaker's broad applicability in gradient-restricted environments, establishing it as a resilient approach for BFAs.*

**Our approach provides a flexible and user-dependent framework, allowing adversaries to choose between gradient-free and gradient-based strategies based on their specific objectives and computational constraints.** For adversaries prioritizing computational efficiency, the gradient-free variant of AttentionBreaker offers a viable solution by leveraging weight magnitudes as a sensitivity proxy, eliminating the need for costly gradient calculations while maintaining high attack efficacy. Conversely, for adversaries willing to accept higher computational costs to minimize perturbations and achieve more precise attacks, the gradient-based version remains advantageous. This adaptability not only enhances the practicality of our method but also strengthens its resilience against existing defenses. Defenses targeting gradient-based attacks may be ineffective against magnitude-based approaches, while those designed to counteract magnitude-based perturbations may fail against gradient-driven strategies. Furthermore, in models with structured dependencies, weight redundancy, or adaptive optimization mechanisms like Adam, gradients play a crucial role in identifying the most impactful perturbations, reinforcing the necessity of gradient-based attacks in certain cases. Given that defenses often redistribute weight significance to mitigate bit-flip attacks, the inclusion of both approaches ensures that our method remains effective across diverse defensive strategies. By offering both computational efficiency and targeted precision, our framework establishes itself as a robust and adaptable bit-flip attack methodology.

### A.5.2 Gray-box AttentionBreaker

While the proposed AttentionBreaker operates as a white-box attack, assuming complete access to model parameters, practical scenarios may restrict access to all parameters. In such cases, grey-box attacks, where only partial model information is available, become pertinent. To assess the robustness of our method under

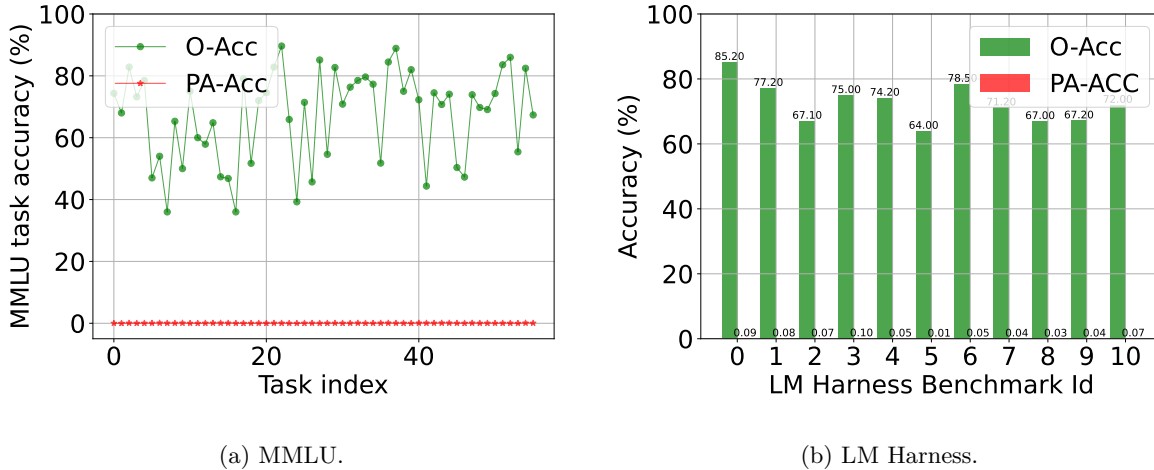

(a) MMLU.

(b) LM Harness.

Figure 9: LLaMA3-8B-Instruct W8 attack transferability analysis across various tasks.

these constraints, we simulate a grey-box scenario to demonstrate the efficacy of the attack. Specifically, we limit accessibility to only a subset of initial model layers.

In order to simulate such limitations during a gray-box attack, we focus our attack strategy exclusively on the initial layers of the model. The initial layers are particularly susceptible to bit-flips, making them ideal candidates for our gray-box attack. Furthermore, even when restricted to having access only to the weight magnitudes, without the gradients, our attack approach remains highly effective, resulting in a significant degradation in model accuracy, from 67.3% to 0%.

### A.5.3 Task Transferability

This experiment examines the transferability of attack effects across distinct tasks. Specifically, it investigates whether an attack executed on Task A leads to observable adverse effects on Task B. If the attack effects indeed transfer between tasks, it would indicate a fundamental vulnerability in the model architecture, independent of the task at hand. Such a finding would underscore the severity of the model's susceptibility to the attack. For this investigation, the LLaMA3-8B W8 model was subjected to a targeted attack on the MMLU task "Astronomy". Due to the attack, the accuracy of the model on "Astronomy" drops from the original accuracy (O-Acc) of 72.1% to a post-attack accuracy (PA-Acc) of 0%. The resulting impact was then systematically evaluated across additional MMLU tasks and other language benchmarks, including the LM harness benchmark. The results reveal a high degree of transferability, whereby the attack yields comparable degradation in performance across the evaluated tasks. This finding underscores the broad vulnerability of the model to such attacks. Figure 9a illustrates that an attack initially targeted at the "Astronomy" task leads to substantial accuracy reductions in other MMLU tasks. Similarly, an attack initially targeted at the "AddSub" task from the LM harness benchmark leads to severe model degradation on other LM harness tasks as well as captured in Figure 9b. *This illustrates the pronounced transferability of the attack across varied tasks and domains.*

### A.5.4 AttentionBreaker Transferability to Fine-tuned models

This experiment evaluates the persistence of adversarial effects from an attack applied to a base model when, after an attack, the model is loaded with previously fine-tuned adapters for a specific task. Specifically, we fine-tune the LLaMA3-8B-Instruct W8 model on the "AddSub" task from the LM harness benchmark and save the resulting fine-tuned adapter. The fine-tuning progress is shown in Figure 10a, denoted as original fine-tuning or Original-FT-Loss. Next, we reload the original base model and perform the AttentionBreaker attack. As anticipated, the model exhibits a substantial loss increase and severe accuracy degradation, with accuracy plummeting from 85.2% to 0%, as illustrated in Figure 10b (labeled as F1). Upon loading the fine-

tuned adapter, we observe that the model accuracy remains at 0%, indicating that *the fine-tuned adapter alone is insufficient to restore model accuracy following the attack.*

### A.5.5   Fine-tuning-based Recovery

This experiment assesses the persistence of adversarial effects introduced through the AttentionBreaker on a base model, specifically evaluating whether subsequent fine-tuning on a task can mitigate these impacts. We begin by attacking the LLaMA3-8B-Instruct W8 model, followed by evaluating the model's accuracy on the "AddSub" task from the LM harness benchmark. As a result, the model's accuracy drops drastically from an initial 85.2% to 0%, as depicted in Figure 10b (labeled as F2), indicating that the model output now consists largely of arbitrary strings devoid of task relevance.

Subsequently, we fine-tune the attacked model on the same task, conducting five epochs with 21 iteration steps per epoch, as shown in Figure 10a. Throughout fine-tuning, the validation loss (or the post-attack fine-tuning loss PostAttack-FT-Loss) declines substantially from an initial 9.12 to 0.27, signaling effective task adaptation. Despite this apparent success in convergence, post-fine-tuning testing reveals that model accuracy remains at 0%, as shown in Figure 10b. This result suggests that while the model now outputs seemingly structured words rather than random strings, it fails to exhibit logical coherence or task-related reasoning, thus remaining unsuccessful in executing the logical deduction necessary for the "AddSub" task. *The findings imply that fine-tuning does not suffice to restore task-specific functionality after the Attention-Breaker attack.*

### A.5.6   Zero-masked fine-tuning

In this study, we investigate the persistence of adversarial effects introduced via the AttentionBreaker method on a base model and evaluate whether task-specific fine-tuning—applied after resetting the perturbed weights to zero—can effectively mitigate these impacts. In particular, we aim to determine whether nullifying the impact of the bit-flipped weights (*i.e.*, setting the perturbed weights to zero, $w_i = 0$) permits effective recovery of model performance through fine-tuning on the same task. We begin attacking the LLaMA3-8B-Instruct W8 model using our AttentionBreaker framework, subsequently evaluating its accuracy on the "AddSub" task from the LM harness benchmark. Immediately following the attack, the model's accuracy plunges from an initial 85.2% to 0%, as illustrated in Figure 10b. Next, we mask the attacked weights by resetting them to zero ($w_i = 0$) and fine-tune the model on the same task, performing five epochs with 21 iterations per epoch. During fine-tuning, the validation loss (or PostAttackZMask-FT-Los) decreases

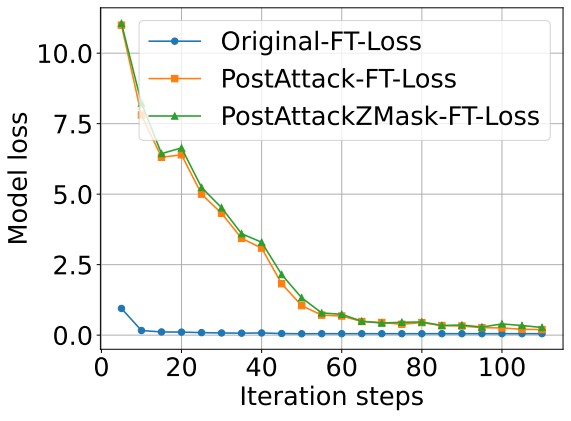

(a) Fine-tuning model loss with iterations.

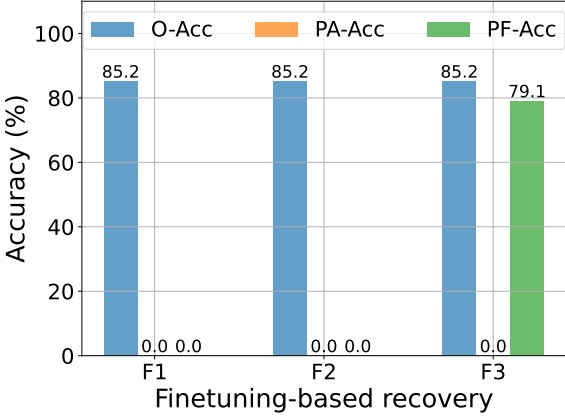

(b) Fine-tuning-based recovery strategies.

Figure 10: Fine-tuning-based recovery after the attack: (a) validation loss with iteration steps and (b) model performance for different fine-tuning strategies. Metrics are calculated on the 'AddSub' task of the LM Harness benchmark.

significantly from an initial value of 9.03 to 0.25, as depicted in Figure 10a, indicating that the model is adapting effectively to the task requirements.

Upon fine-tuning, we test the model performance on the same task and observe that the model's accuracy is substantially recovered from 0% to 79.1%, as shown in Figure 10b (labeled as F3). *These findings suggest that fine-tuning, preceded by zeroing out perturbed weights, is sufficient to restore task-specific functionality, thereby mitigating the adversarial effects induced by the attack.*

