# OpenReview forum: "AttentionBreaker: Adaptive Evolutionary Optimization for Unmasking Vulnerabilities in LLMs through Bit-Flip Attacks"
_TMLR — Accepted by TMLR_

### Review · Reviewer_gKNh · 2025-07-24

**Summary Of Contributions:**

This paper presents `AttentionBreaker`, a novel and efficient Bit-Flip Attack (BFA) framework for LLMs. The authors demonstrate that merely 3-bits perturbation can collapse MMLU accuracy of `LLaMA3-8B-Instruct` from 67.3% to 0%, which also challenge existing state-of-the-art defenses.

Key contributions include:

- `AttentionBreaker` Framework: An approach combing layer sensitivity ranking and weight-set optimization to efficiently identify critical parameters in LLMs for BFA.
- `GenBFA` algorithm: A novel evolutionary strategy to optimize the selection of minimal yet impactful subset of weights, achieving a 2000x reduction in the number of required bit-flips compared to exhaustive search.
- Quantization-Aware Evaluation: Demonstrates the effectiveness of `AttentionBreaker` across INT8, INT4, and 1.58-bit quantized models.
- Multi-model Extension: Extends the attack to VLMs like `LLaVA1.6-Mistral-7B` , achieving zero-shot task degradation with just 15 flips.

**Audience:**

Yes

**Broader Impact Concerns:**

This paper exposes vulnerabilities in LLMs which could be exploited maliciously to degrade deployed models. The paper should:

- warn about dual-use risks(e.g. attacks via shared hardware)
- discuss regulatory implications(e.g. for hardware-level protections in `MLaaS`)

**Claims And Evidence:**

Yes

**Requested Changes:**

Important

- Add a dedicated section discussing ethical implications, including potential misuse
- Experimentally evaluate the proposed XOR-cipher defense against `AttentionBreaker`
- Discuss hardware constraints in real-world and validate attacks under noisy bit-flip conditions

Strengthen

- report runtime and memory costs for larger or different architecture models
- compare alternative defenses beyond just `FaR`

**Strengths And Weaknesses:**

Strengths:

- First work to demonstrate a 3-bit BFA can collapse a 8B-level LLM, overcoming the limitation of iterative gradient recalculations.
- Rigorous methodology combing layer-wise sensitivity scoring, hybrid gradient-magnitude metrics, and evolutionary weight-subset optimization.
- Covers diverse models(`LLaMA3`, `Phi-3`, `LLaVA`) and quantization schemes (INT8, INT4, 1.58bit).
- Highlights real-world risks for LLMs deployed on shared hardware.

Weaknesses:

- The evolutionary optimization may not scale so efficiently to even more larger models (e.g., 70B+ parameters) or some other `MoE` architectures.
- The method relies on idealized bit-flip models, without accounting for real-world noise or error correction.
- The proposed XOR-cipher defense is not experimentally validated, leaving effectiveness unclear.
- While a defense is proposed, the paper still lacks a broader i

---

> ### Author Response · Authors · 2025-08-15
> **We added dedicated sections on ethical implications, dual-use risks, and regulatory considerations in Section 8 of the revised manuscript. We experimentally evaluated the proposed XOR-cipher/logic-locking defense. We validated attack robustness under noisy bit-flip conditions.**
>
> **Response regarding ethical implications, including potential misuse**
>
> We have added a dedicated Section 8 in the revised manuscript to discuss ethical implications, dual-use risks(e.g. attacks via shared hardware) and regulatory implications(e.g. for hardware-level protections in MLaaS) regarding the proposed AttentionBreaker.
>
>
> **Response regarding experimental evaluation of the XOR-cipher defense**
>
> To experimentally validate the proposed XOR-cipher/logic-locking–based defense, we apply XOR-locking to the vulnerable bits identified by AttentionBreaker, ensuring that only the correct key restores baseline accuracy, while any incorrect key leaves the model in a severely degraded state, thereby deterring further attacks. The security scales exponentially with key length; for example, locking 100 bits with a 100-bit key yields $2^{100}$ possible combinations, making brute-force infeasible. To harden protection when the critical subset is small (e.g., 3 bits), we lock a larger superset. In one experiment with LLaMA3-8B-W8, locking 100 bits restored 67.3% MMLU accuracy with the correct key, but incorrect keys reduced accuracy to an average of 26.7%, often causing complete accuracy collapse (62% cases). Furthermore, once an incorrect key degrades the model, any subsequent AttentionBreaker attack has low incentive and efficiency, as the optimization process targets performance that is already severely impaired. Consequently, without the correct key, the model becomes functionally unusable, providing no incentive for follow-up adversarial attacks.
>
>
> **Response regarding attack validation under noisy bit-flip conditions**
>
>
> We thank the reviewer for highlighting the importance of real-world hardware constraints and noisy bit-flip conditions. In practice, bit-flip injections may be imperfect due to factors such as ECC, and other fault-tolerance mechanisms in modern accelerators. To evaluate this, we performed a preliminary noisy bit-flip experiment after identifying the critical bits targeted by AttentionBreaker. Each intended flip was applied with probability p (0.8–1.0), while unintended flips within the same parameter tensor occurred with probability q (0.0–0.3). The choice of p and q values are primarily system dependent and p near 1 and a non-zero q is an acceptable assumption. Each configuration was executed 100 times and averaged. The results, presented in model loss/model accuracy format in the table, indicate that even under high noise conditions, the attack remains effective. Specifically, over 82% of cases resulted in full accuracy collapse (attack loss > 7 on the LLaMA3-8B-Instruct W8 model, MMLU benchmark), and all cases exhibited a relative accuracy drop exceeding 50%. These findings demonstrate that AttentionBreaker maintains strong efficacy under realistic noisy bit-flip conditions.
>
>
> |Flip probability of the targeted bit (p)|0.8|0.85|0.9|0.95|1.0|
> |---|--|---|--|---|--|
> |**Flip probability of an untargeted bit(q)**||||||
> |**0.0**|7.38/0| 9.51/0| 9.9/0|11.4/0|13.1/0|
> |**0.05**|7.11/0.3|7.48/0.2|9.0/0|10.8/0|12.7/0|
> |**0.1**|8.15/0|8.8/0|8.1/0|10.6/0| 11.8/0|
> |**0.15**|7.23/0|8.9/0|8.5/0|11.1/0|11.4/0|
> |**0.2**|5.33/21.4|5.8/14.8|8.4/0|10.1/0|10.2/0|
> |**0.25**|6.0/15.3|7.1/2.2|7.9/0|8.6/0| 9.9/0|
> |**0.3**|3.83/21.3| 6.41/5.3|6.59/1.2|8.13/0|9.62/0|
>
>
> **Response regarding runtime and memory costs**
>
> We thank the reviewer for this suggestion. As detailed in Section 5.3.3 of the paper, for LLaMA3-8B-Instruct-W8, AttentionBreaker identifies critical bit-flips in approximately 2 hours on 4×A100 GPUs, requiring around 80 GB of memory. We extended this evaluation to a larger model, MiniMixtral-v0.2 (12.9B parameters), where the process required ~127 GB of memory and 2.2 hours to converge and identify the most critical bits.
>
>
> **Response regarding alternative defenses beyond FaR**
>
> To the best of our knowledge, FaR is the only defense specifically targeting bit-flip attacks on transformer-based models and we show its ineffectiveness against AttentionBreaker in Section 6 of the paper. Nonetheless, we discussed our proposed attack against state-of-the-art DNN defenses in Section 6 of the paper. Aegis [1] uses multi-exit classifiers with random selection, but fails if early layers are corrupted, which is an attack strategy AttentionBreaker exploits effectively. Similarly, low bit-width quantization [2], proposed to reduce numerical precision to limit bit-flip impact, also fails as AttentionBreaker is effective even in quantized models. For instance, on LLaMA3-8B-Instruct (NF4, 4-bit), our method reduces accuracy from 60.3 % to 0 % with only 28 bit-flips. These findings underscore the need for LLM-specific protections.
>
> [1] Wang, Jialai, et al. “Aegis: Mitigating targeted bit-flip attacks against deep neural networks.” USENIX, 2023.
>
> [2] He, Zhezhi, et al. “Defending and harnessing the bit-flip based adversarial weight attack.” CVPR, 2020.

---

> ### Comment · Reviewer_gKNh · 2025-08-16
> **Fundamental issue with evolutionary algorithms**
>
> The key issue with the proposed method is that it relies on evolutionary algorithms, which cannot effectively handle high-dimensional problems. The authors did not respond to my question "The evolutionary optimization may not scale so efficiently to even more larger models (e.g., 70B+ parameters) or some other MoE architectures."

---

> ### Author Response · Authors · 2025-08-17
> **We addressed the scalability of the evolutionary optimization to larger models (e.g., 70B+ parameters) or MoE architectures**
>
> **Response regarding the scalability of the evolutionary optimization to larger models (e.g., 70B+ parameters) or MoE architectures**
>
> We sincerely thank the reviewer for raising this important concern. We would like to respectfully note that, based on our understanding of the reviewer’s initial comments under the “requested changes” column, the question regarding scalability to larger models (e.g., 70B+ parameters or MoE architectures) was not explicitly raised, and therefore we prioritized addressing the requested changes. Nonetheless, we are happy to provide clarification and additional experimental results to demonstrate the broader applicability of our method.
>
> First, we evaluated AttentionBreaker on a Mixture-of-Experts (MoE) architecture. In the MiniMixtral-v0.2 model (12.9B parameters, BFloat16 precision), only two carefully chosen bit-flips were sufficient to cause catastrophic collapse, reducing MMLU accuracy from 58.55% to 0 and increasing WikiText-2 perplexity from 20.78 to 2.1×10⁴. This demonstrates that evolutionary optimization remains effective even in more complex architectures such as MoE.
>
> Second, we fully agree that scaling experiments to 70B+ parameter models is critical. However, due to resource limitations in an academic environment (4× NVIDIA A100 GPUs with 320 GB total memory), we cannot feasibly compute full gradient-based sensitivity for models of this scale, as training or gradient evaluation would exceed our compute budget. To still address this point constructively, we performed an alternative magnitude-based sensitivity analysis on the LLaMA 3.3 70B Instruct model in 4-bit NF4 quantization. Using magnitude-based sensitivity scoring (as discussed in Section A.5.2 of the paper), we collected the initial subset of critical weights of size 23488. Following this we employed our GenBFA evolutionary optimization to reduce the most critical subset of bits and found that flipping only a **single** critical bit was sufficient to reduce MMLU accuracy from 86% to 0 and increasing WikiText-2 perplexity from 9.53 to 1.2×10⁴. This result demonstrates that AttentionBreaker scales to very large models and maintains high attack efficacy.
> | Model                  | Model type | Precision   | Parameters | # Bit-Flips | WikiText-2 Perplexity (Before/After) | MMLU Accuracy (%) Before/After attack |
> |-------------------------|------------|-------------|------------|-------------|--------------------------------------|----------------------------------|
> | MiniMixtral-v0.2        | LLM (MoE)  | BFloat16    | 12.9B      | 2           | 20.78 / 2.1×10⁴                      | 58.55 / 0                        |
> | LLaMA 3.3 70B Instruct  | LLM        | NF4 (4-bit) | 70.6B        | 1          | 9.53 / 1.2×10⁴                                | 86.0 / 0                       |
>
> We hope these additional results provide reassurance that AttentionBreaker’s evolutionary optimization is not only effective for models up to 13B parameters, including MoE architectures, but also remains practical for large-scale models beyond 70B parameters.

---

### Review · Reviewer_7LX8 · 2025-07-25

**Summary Of Contributions:**

This paper introduces a novel bit-flip attack method called AttentionBreaker, which significantly reduces the performance of large language models (LLMs) across multiple public datasets. AttentionBreaker identifies the most vulnerable weights in a neural network—those that have the greatest impact on the model's output—and flips their most significant bits (MSB) to create corrupted weights. The algorithm employs a scanning and combinatorial optimization mechanism to inflict maximum damage on the LLM's performance while minimizing the number of flipped bits.

AttentionBreaker operates in two stages. In the first stage, the layer ranking module ranks weights within each layer by calculating a sensitivity score, which is a convex combination of their magnitude and gradients. The top-k most sensitive weights are selected for bit-flipping, where k is determined based on the layer's size. A layer score is then calculated based on the loss increase caused by modifying these sensitive weights, and all layers are ranked according to this score.

In the second stage, GenBFA, a heuristic evolutionary algorithm is used to further refine the set of perturbed weights, maximizing the damage to the model while minimizing the number of perturbed weights.

Empirical studies conducted on multiple public datasets across various open-source LLM architectures with different quantization schemes demonstrate the effectiveness of the proposed algorithm. Ablation studies explore different quantization levels, bit-flip budgets, evolutionary algorithms, and hyperparameters. The paper also examines generalization to gray-box attacks and the transferability of attacks to fine-tuning scenarios, with qualitative examples provided.

**Audience:**

Yes

**Claims And Evidence:**

Yes

**Requested Changes:**

1. Comparison with Baseline Using DeepHammer: Evaluate the performance of the proposed method against a baseline where DeepHammer is applied to the initial, final, or a random layer.  For e.g., The bit sensitive score of DeepHammer could be modified to maximum the loss in order to accommodate un-targeted attack. This comparison will aid in assessing the significance of this work's contribution.

2. Compare the results to a baseline where the top three most sensitive weights in the most sensitive layer are corrupted, to justify the necessity of using the evolutionary algorithm.

3. Present the attack results on other popular open-source models such as Qwen, Mixtral, and DeepSeek.

4. Show the performance on full-precision scenario if possible

**Strengths And Weaknesses:**

Pros:

1. This work introduces a tool that effectively scans and identifies sensitive weights within large language models (LLMs), offering valuable insights for other areas such as model compression and Trojan attack/defense.

2. The empirical study demonstrates significant improvements over existing baselines (Nazari et al., 2024). By flipping just three bits in an 8-billion parameter model, the proposed method can completely reduce the LLM's accuracy to zero.

Cons:

1. From my perspective, the primary distinction between this work and the existing method, DeepHammer, as mentioned in the paper, is the model size they are facing. DeepHammer is challenging to extend to the LLM setting due to its exhaustive nature, making it costly to attack LLMs. A simple baseline could involve applying DeepHammer only to the initial or last layer. The bit sensitive score of DeepHammer could be modified to maximum the loss in order to accommodate un-targeted attack.

2. Some ablation studies are still missing. For instance, what would happen if we simply corrupted the top three most sensitive weights in the most sensitive layer? Alternatively, what if we only corrupted the top three most sensitive weights in the initial or output layer without using the complex evolutionary algorithm?

3. While I understand that quantized models present a more challenging setting for bit-flip attacks, it would be beneficial to evaluate the attack's performance on models in their original precision. The intuition here is that full-precision models might be more capable of handling errors due to their additional capacity.

4. The empirical study is heavily focused on the LLaMA series models. It would be advantageous to see attack results on other popular open-source models like Qwen and Mixtral.

---

> ### Author Response · Authors · 2025-08-15
> **We compared AttentionBreaker with DeepHammer and found it to be significantly more efficient. We demonstrated the necessity of our evolutionary algorithm. We showed further evaluations on diverse open-source models, in full-precision settings.**
>
> **Response regarding comparison with DeepHammer**
>
> Although DeepHammer was originally proposed for DNNs, we evaluated its performance alongside AttentionBreaker on the Qwen3-0.6B-Base model by targeting the initial layer, a randomly selected layer, and the final layer. The results show that AttentionBreaker is substantially more efficient when attacking the initial and final layers, consistent with our analysis in Section 5.3.3 of the paper. Specifically, AttentionBreaker required only 5 and 3 bit-flips, respectively, to cause severe degradation when targeting the initial and final layers, compared to 4,687 and 3,325 bit-flips for DeepHammer, which is approximately 1000 times more.
> | Model |Layer type | DeepHammer # Bit-Flips Required | AttentionBreaker # Bit-Flips Required|MMLU accuracy before/after attack (%)|
> |--|--|--|--|--|
> | Qwen3-0.6B-Base|Initial| 4,687| 5|49.6/0|
> | Qwen3-0.6B-Base|Random|2,022|1954|49.6/0|
> | Qwen3-0.6B-Base | Final|3,325| 3|49.6/0|
>
> **Response regarding the necessity of the Evolutionary algorithm**
>
> As shown in Section A.2 in the paper, when using a sensitivity ratio of 0.5, a naive bit selection strategy and even when targeting the most critical bits, produces negligible accuracy degradation. For example, in LLaMA3-8B-W8, flipping the top three most sensitive bits in the most sensitive layer without optimization leaves MMLU accuracy at 66.2%. In contrast, GenBFA’s evolutionary optimization with the same three flips reduces accuracy from to 0.0%, a complete collapse. This dramatic improvement holds across all evaluated models. These results clearly show that bit importance is non-additive and simply ranking bits by sensitivity is insufficient and that the evolutionary algorithm is essential for identifying minimal yet jointly critical subsets of bits that maximize the attack’s effectiveness.
>
> | Model| # Bit-Flips (Pre-Optimization) | MMLU Accuracy (%) | # Bit-Flips (Post-Optimization) | MMLU Accuracy (%) |
> |--|---|--|---|--|
> | LLaMA3-8B-Instruct-W8| 3|66.2|3|0.0|
> | LLaMA3-8B-Instruct-W4|28|58.3| 28 | 0.0|
> | Phi-3-mini-128k-Instruct-W8|4|68.9|4|0.0|
>
> **Response regarding attack efficacy on other open source models**
>
> We thank the reviewer for suggesting an evaluation of our method on additional open-source models beyond those reported in the original work. The following table presents the results of applying AttentionBreaker to various full-precision (BFloat16) LLMs not covered in the initial submission. For example, in the Deepseek-7B model, flipping only four critical bits increased WikiText-2 perplexity from 9.92 to 4.2×10⁴ and reduced MMLU accuracy from 41.8% to 0%. Similarly, in Qwen3-0.6B-Base, a single bit-flip was sufficient to induce catastrophic performance collapse, while in the much larger MiniMixtral-v0.2 (12.9B parameters), only two bit-flips caused WikiText-2 perplexity to spike from 20.78 to 2.1×10⁴ and MMLU accuracy to drop from 58.55% to 0%. This confirms the universal vulnerability of LLMs to bit-flip attacks.
> |Model|Model type|Precision|Number of  Parameters|AttentionBreaker (# of bit-flips)|Benchmarks (Before attack / After Attack)||
> |:---:|:---:|:---:|:---:|:---:|:---:|:---:|
> ||||||WikiText-2 Perplexity|MMLU (Accuracy in %)|
> |Deepseek-7B|LLM|BFloat16|7B|4| 9.92/4.2e4|41.8/0|
> |Qwen3-0.6B-Base|LLM|  BFloat16|596M |1|18.23/3.2e4|49.6/0|
> |MiniMixtral-v0.2|LLM|BFloat16|12.9B|2|20.78/2.1e4|58.55/0|
>
> **Response regarding attack efficacy on full precision model**
>
> We thank the reviewer for the suggestion to present results on full-precision models. The following table demonstrates the efficacy of AttentionBreaker when applied to full-precision (BFloat16) models, showing that it can be even more effective than in quantized settings. This increased efficiency arises from the larger representational range of full-precision formats compared to quantized values, which amplifies the effect of targeted bit-flips, as discussed in Section A.1.3 in the paper. Across diverse model architectures AttentionBreaker achieves complete accuracy collapse with as few as 1–3 bit-flips. For instance, in LLaMA-3, flipping just two strategically chosen bits increased WikiText-2 perplexity performance from 10.24 to 3.8×10⁴ and MMLU accuracy reduced from 66.3% to 0%. These results highlight that even minimal bit-flips in full-precision models can induce catastrophic performance degradation.
> | Model|Model type|Precision|Number of  Parameters|AttentionBreaker (# of bit-flips)|Benchmarks (Accuracy in %)(Before attack / After Attack)||||
> |:--:|:--:|:--:|:--:|:--:|:---:|:---:|:---:|:--:|
> ||||||WikiText-2 Perplexity|MMLU |VQAv2|TextVQA |
> |LLaMa-3 Instruct|LLM|BFloat16| 8.03B| 2|10.24/3.8e4|68.4/0|NA|NA|
> |LLaVa mistral 1.6|VLM|BFloat16|7.57B|1|NA |NA|77.2/0|63.6/0|
> |Phi-3-mini-128k Instruct|LLM|BFloat16|3.82B|3 |9.45/3.6e4|68.1/0|NA|NA|
> |LLaMa-3.2|LLM|BFloat16|1.24B |2|20.78/2.1e4| 49.3/0|NA|NA|
> |Qwen3-0.6B-Base|LLM|BFloat16|596M|1|18.23/3.2e4|49.6/0|NA|NA|

---

> > ### Comment · Reviewer_7LX8 · 2025-08-15
> > **Response to authors' reply**
> >
> > Thank you to the authors for their efforts in addressing my concerns. The response resolves most of my questions. It would be beneficial to include the final accuracy after the attack in the DeepHammer experiment (Table 1 in the response) in your final version.

---

### Review · Reviewer_3zHX · 2025-08-02

**Summary Of Contributions:**

The paper proposes an efficient bit-flip attack on large language models. While the previous work requires a large number of flipped bits to make the attack effective, the proposed method only needs a significantly smaller amount. The proposed method first selects the most sensitive layers and weights with a sensitive score combining the weight and gradient magnitude. An evolution-based method is then proposed to reduce the size of the selected weights while maintaining a good attack performance. Experiments have been conducted on several open-sourced language models and show that a very small number of bits could lead to a significant performance degradation.

**Audience:**

Yes

**Broader Impact Concerns:**

None.

**Claims And Evidence:**

Yes

**Requested Changes:**

Please refer to the cons. The paper needs a major revision on the clarity.

**Strengths And Weaknesses:**

Pros:
1. The proposed method shows a great improvement in using significantly fewer bits to achieve a great attack performance.

Cons:
1. The proposed method doesn't specify the threat model. It didn't specify what loss is used and how the loss will be achieved. Since the attack is on the language model, the loss should be different from the classification task. I am not sure whether the loss is applied with every downstream task or just the pretraining stage. Also, I am not sure if the loss is obtained from the training datasets or the test datasets.
2. The proposed method lacks clarity. Some key details are without introduction and make the paper hard to follow. The paper introduces how to find the most sensitive weights, but I don't get how to find the most effective bits from the weights. Will the bit on the left always be selected, or is the pos the parameter to control the selection? The GenBFA also shows similar issues. What is the gene g in the crossover operation, and how to mutate the two subsets? I can get the rough picture, but there are not enough details.
3. The methodology section needs a major revision of the algorithm description. There is no need to explain every line of an algorithm, such as storing some parameter somewhere. Functions such as SScore can be easily replaced by equation 1.
4. The paper's attempt at a targeted case is interesting; however, I don't see how the method can be adapted. It could also be related to the first con that the loss function is never clearly stated.

---

> ### Author Response · Authors · 2025-08-17
> **We clarified the threat model, bit selection, GenBFA details and loss formulation in targeted attack case.**
>
> **Response regarding the threat model**
>
> We thank the reviewer for raising these important points, which have helped us clarify and strengthen the methodology section. The threat model is defined in Section 4.1 as a white-box setting in which the adversary has full access to both the model architecture and parameters. Our attack is primarily untargeted, aiming to degrade overall model performance across downstream tasks. To evaluate this, we use the standard cross-entropy loss between the model output logits and the encoded input token IDs. Importantly, this loss is evaluated independently for each downstream benchmark (for example, MMLU in the case of LLMs) using the respective task-specific test datasets rather than training data. This ensures that the measured degradation reflects realistic evaluation conditions. We have revised Section 4.1 to make this threat model and the loss definition fully explicit.
>
> **Response regarding clarity on bit selection and GenBFA**
>
> We thank the reviewer for raising these concerns. As described in Section 4.2.1, we initially allowed arbitrary bit selection within weights through a variable parameter. However, experimental evidence consistently demonstrated that flipping the most significant bit (MSB)—the left-most bit in the quantized representation and the sign bit in full precision—induces the largest deviation from the original value and maximizes model loss and attack effectiveness. Accordingly, our final method always selects the MSB, and this is now explicitly stated in the manuscript. With respect to GenBFA, each candidate solution, or “gene,” represents a subset of weight indices to be flipped. The crossover operation combines solutions probabilistically: for each bit index, inheritance occurs either from the best-performing solution of the current generation or from a randomly selected parent, as formalized in Equation (9). Mutation is performed by randomly reducing the size of the candidate solution (weight subset) according to a variable reduction rate (ranging between 1–10% and chosen randomly during each mutation call), thereby introducing evolutionary pressure variably toward compact, high-impact solutions. The number of generations is capped at 100, with early stopping applied whenever no improvement is observed for a fixed number of iterations (e.g. 5). Section 4.2 has been revised to clearly articulate these design choices and operations.
>
> **Response regarding the algorithm description**
>
> We thank the reviewer for suggesting improvements to the algorithm presentation. We have also substantially revised Section 4.2.4. The original line-by-line implementation details, have been removed, and functions such as SScore have been replaced with direct references to the relevant equations (e.g., Equation 1). The revised description now emphasizes only the essential stages of the procedure such as crossover, mutation, scoring, and termination and thus providing clarity and conciseness.
>
> **Response regarding the targeted attack case and loss function**
>
> We are grateful for the reviewer’s interest in the targeted attack extension. We have strengthened the discussion of the targeted attack case. In this setting, the attack objective is not to degrade model performance globally but to force targeted misclassification. Towards this, let us discuss a 1-to-1 targeted misclassification attack on the MMLU benchmark ``Astronomy'' task using the LLaMA-3.2-1B-Instruct-W8 model. The goal was to cause inputs originally belonging to Class A to be intentionally misclassified as Class B.
> The experiment involved four classes (A, B, C, D), with model accuracy and targeted misclassification rates measured before and after the attack. In this scenario, when the true class is A, the attack loss $L_{attack}$ is defined as the negative log-likelihood of predicting class B; minimizing this term directly encourages the model to misclassify A as B. To ensure that the attack remains selective rather than indiscriminate, we introduce a correctness loss $L_{correct}$ applied to all other classes (B, C, and D in the MMLU example). This term is defined as the negative log-likelihood of predicting the correct class label, and it enforces that normal accuracy is preserved outside of the A→B mapping. The total loss is formulated as the sum of the attack loss and the correctness loss, weighted by a tunable hyperparameter λ (defaults to 0.5) that balances attack strength against preservation of normal behavior. Minimizing this total loss ensures that the targeted misclassification is achieved while predictions on non-target classes remain accurate. Section 5.3.4 has been revised to clearly define this formulation and its role in enabling targeted attacks under our framework.

---

### Decision · Action_Editor_up7r · 2025-10-15

**Recommendation:** Accept with minor revision

**Audience:**

Yes

**Audience Explanation:**

yes. The problem is timely and could be of interest to a broad audience interested in AI safety and in LLM.

**Claims And Evidence:**

Yes

**Claims Explanation:**

This paper proposes an interesting bit-flip attack targeting modern Large Language Models (LLMs). The core methodology combines a heuristic evolutionary algorithm with a layer selection strategy. Extensive experiments demonstrate the efficacy of the proposed method. A significant amount of work has been done to revise the manuscript and address initial reviewer concerns.

Overall, reviewers commended the paper's novelty and the extensive experiments. However, some concerns were not fully addressed during the rebuttal, specifically those revolving around the threat model, the choice of loss function, and scalability.

While these are valid concerns, they will require singificant changes that would fundamentally demand a new paper: shifting the focus from post-training to pre-training, extending the loss from classification to sequence generation, or requiring experiments on extremely large models that are often inaccessible to academic labs. Overall, this Area Chair finds the work to be already strong and solid, and worth publish.